# A globally consistent negative effect of edge on aboveground forest biomass

Gayoung Yang [1,2] ✉, Thomas W. Crowther [1], Thomas Lauber [1], Constantin M. Zohner [1] & Gabriel Reuben Smith [1]

Because of widespread forest fragmentation, 70% of the world's forest area lies within 1 km of an edge. Forest biomass density near edges often differs markedly from biomass density in the interior. In some biomes, these 'edge effects' are responsible for substantial reductions in forest carbon storage. However, there is little consensus on the direction and magnitude of edge effects on forest biomass across the globe, which hampers their consideration in forest carbon stock accounting. Here we examined eight million forested locations to quantify variability in edge effects on biomass at a global scale. We found negative edge effects across 97% of examined areas, with aboveground biomass density on average 16% lower near edges than in interior forests. Higher temperature, precipitation and proportion of agricultural land were linked to more negative edge effects. Along with differences in the spatial scale of analysis, this variation can explain contrasting observations among previous studies. We estimate that edge effects have reduced the total aboveground biomass of forests by 9%, equivalent to a loss of 58 Pg. These findings underscore the substantial impact of forest fragmentation on global biomass stocks and highlight the critical need to account for edge effects in carbon stock assessments.

Forests are a major carbon sink[1], sequestering ~24% of annual anthropogenic carbon emissions[2,3]. However, these ecosystems are under severe threat[4], with 420 million ha of forest lost globally between 1990 and 2020, leading to substantial reductions in carbon storage[5]. Contiguous forest landscapes are increasingly fragmented[6], dividing large tracts into smaller patches and creating new edges along their perimeters. Currently, 30% of the world's forested areas lie within 100 meters of an edge, and 70% are situated less than a kilometre away[7]. This fragmentation introduces 'edge effects', characterized by gradual changes in biodiversity[7,8], microclimate[9,10], soil conditions[11] and exposure to human influences[12–14], such as nutrient inputs and selective logging, from the forest edge to the interior. These changes can directly influence the growth potential and carbon storage capacity of forests[7,9]. As fragmentation accelerates[15], understanding the scale and implications of these 'edge effects' is becoming critical for predicting terrestrial carbon storage under current and future climate scenarios.

Near forest edges, more solar radiation can usually penetrate the canopy, driving increases in air and soil temperatures, which in turn elevate vapour pressure deficit (VPD) and decrease soil moisture[9]. Forest edges also typically experience stronger wind exposure, and are more vulnerable to fire[16] and biotic disturbances such as invasive species[17]. These microenvironmental gradients between forest edges and interiors are expected to influence vegetation growth and biomass density. Yet, empirical evidence regarding the sign and magnitude of edge effects on forest biomass across the globe is mixed. In tropical regions, biomass density has been shown to decrease near forest edges[18,19]. In temperate forests, the effects are highly variable, with studies showing positive, negative, or negligible impacts in different regions[6,8,20–23]. In boreal forests, both higher vegetation productivity and higher tree mortality have been observed along edges, leading to contrasting impacts on total forest biomass[6,24].

[1]Institute of Integrative Biology, Department of Environmental Systems Science, ETH Zürich, Zurich, Switzerland. [2]Sorbonne Université, Paris, France. ✉e-mail: gayoung.yang@usys.ethz.ch

The lack of consensus on the biogeographic patterns of edge effects in different ecoregions limits our capacity to represent them in carbon stock accounting efforts[6,19,20,25]. Current carbon accounting efforts thus generally overlook these indirect effects of forest fragmentation[6], focusing instead on the direct effects of absolute forest loss on carbon stocks[5,26]. For example, the Tier 1 methodology of the Intergovernmental Panel on Climate Change instructs countries to estimate greenhouse gas inventories using fixed per-hectare carbon stock values for each forest type, without differentiating between edge and interior areas[19,27]. Where edge effects are pronounced, this approach risks substantial over- or underestimation of actual carbon stocks[19], hampering effective carbon stock assessments and climate change policy[28,29].

Here we address this issue by empirically quantifying the relationship between aboveground forest biomass (AGB) and distance from forest edges on a global scale. Recognizing that various mechanisms influence biomass near edges, we define edge effects as the net outcome of all factors shaping biomass variation along the edge–interior gradient. To better understand the potential drivers behind this variation, we employ interpretable machine-learning techniques to identify key environmental and anthropogenic contributors. Finally, we estimate the total global impact of edge effects on forest AGB.

## Results and discussion
### Global variation in edge effects
To measure forest edge effects at a global scale, we combined the high-resolution (30 m) global forest cover map from ref. [30] with the high-resolution (30 m) global forest biomass map created from ref. [31]. We overlaid a 100 km × 100 km grid across the global forest area and sampled 500 random points within each grid cell. The spatial distribution of these points followed that of global forested areas (Fig. 1b,c). Using these sampled points, we fit spatial log-linear regression models at the individual grid cell level, predicting biomass density as a function of the $\log_{10}$-transformed distance to the forest edge while accounting for spatial autocorrelation (Methods). The resulting slopes, denoted as $\frac{\Delta \text{AGB}}{\Delta D}$, represent the local relationships between forest biomass and distance to edge $D$ within each grid cell.

We found that most (96.1%) grid cells displayed $\frac{\Delta \text{AGB}}{\Delta D} > 0.1$ (Fig. 1a; see Extended Data Fig. 1 for uncertainty). The mean $\frac{\Delta \text{AGB}}{\Delta D}$ was positive across all forest biomes (Fig. 1d), indicating negative edge effects, where biomass density near edges is consistently lower than in interior forests. Tropical forests exhibited the strongest negative edge effects, particularly in regions such as Southeast Asia, the Amazon, Central America and the Congo Basin (Fig. 1a,d). Temperate forests exhibited a 19% lower $\frac{\Delta \text{AGB}}{\Delta D}$ compared with tropical forests (mean $\frac{\Delta \text{AGB}}{\Delta D}$ = 53 for tropical and 43 for temperate forests, respectively, Fig. 1d). Nevertheless, we observed strong negative edge effects in temperate regions such as Europe and the United States. In boreal forests, we observed weaker negative edge effects, except for the Western Siberian grain belt in Russia, where strong negative edge effects were observed.

Positive edge effects ($\frac{\Delta \text{AGB}}{\Delta D} < 0$) accounted for only 3.7% of the total observed values and were primarily restricted to regions near the biophysical growth limits of trees, such as high-latitude boreal forests. A negligible edge effect ($\frac{\Delta \text{AGB}}{\Delta D}$ between −0.1 and 0.1) was observed in only 0.2% of grid cells.

To confirm the robustness of our results, we conducted supplemental analyses. First, to ensure that the statistical method does not bias the results, we replaced log-linear regression with non-parametric Spearman correlations, which produced qualitatively similar results (Extended Data Fig. 2a). Second, we excluded points within 30 m of forest edges to ensure that mixed pixels at the border of forests were not

driving the observed patterns. This analysis also returned consistent results (Extended Data Fig. 2b). Lastly, to address potential inaccuracies in the aboveground forest biomass map, we repeated the analysis using tree canopy cover[30] as a response variable instead of biomass. We found analogous results (Extended Data Fig. 3), suggesting that reduced biomass near edges is most probably attributable to decreases in canopy cover rather than data artefacts.

### Environmental drivers of edge effects
To identify the environmental factors underlying global variation in edge effects, we combined an Extreme Gradient Boosting (XGBoost) machine-learning model[32] and Shapley Additive Explanation (SHAP) values[33] (Methods). Because edge effects were already quantified at the grid level using spatial log-linear regressions (Fig. 1), the purpose of this machine-learning analysis was not to generate new predictions, but rather to interpret which environmental variables contribute most to observed variation in edge-effect magnitude. SHAP values are particularly well suited for this task, as they quantify the contribution of each variable to the model's estimation of $\frac{\Delta \text{AGB}}{\Delta D}$ for a given grid cell.

The mean |SHAP| value indicates the overall importance of a variable across all locations. For example, if a variable has both high values and high positive SHAP values—as observed for agricultural land cover in Fig. 2—this suggests that areas with extensive agriculture tend to exhibit stronger negative edge effects. In contrast, if high values of a variable are associated with negative SHAP values, it implies that the variable tends to suppress the magnitude of the edge effect.

To propagate uncertainty in our grid-cell estimates of $\frac{\Delta \text{AGB}}{\Delta D}$, we weighted each estimate by the inverse of its coefficient of variation. This approach, commonly used in meta-analyses, gives greater emphasis to effect size estimates with lower uncertainty[34]. To ensure that spatial patterns did not bias our evaluation of machine-learning model performance, we used a spatially buffered leave-one-out cross-validation approach to calculate $R^2$ for our models (Extended Data Table 1). Using this method, we developed both global and biome-specific models (Fig. 2 and Extended Data Fig. 4), selecting environmental predictors on the basis of previous literature (Extended Data Table 2) and confirming no multicollinearity (Spearman's rank correlation coefficients <0.7 and variance inflation factors (VIFs) <3 for all predictors).

Our global-scale machine-learning model had an $R^2$ of 0.67. Among the predictors, mean annual temperature (MAT) was the most important variable, with a mean |SHAP| value of 7.2, followed by the percentage of cultivated and managed vegetation (Agriculture in Fig. 2) (4.9) and mean annual precipitation (MAP) (3.9) (Fig. 2a).

In colder regions such as boreal forests, temperature is the limiting factor for plant growth[6]. In these areas, higher temperatures near forest edges during summer months can promote vegetation growth during the growing season[35], resulting in a negative SHAP value for low MAT (Fig. 2b). In contrast, low temperature is generally not a limiting factor in tropical forests. In these biomes, higher temperatures near the edge may instead increase the vulnerability of trees to heat stress during the growing season[6], causing a positive SHAP value in regions with high MAT (Fig. 2b).

A high fraction of agriculture was identified as the second most important variable negatively impacting global edge biomass (Fig. 2a,c), aligning with the findings of ref. [19] in the tropics. Fires near edges, often driven by agricultural expansion, have been identified as a key driver of Amazon forest fragmentation and degradation, a process exacerbated during droughts[36,37]. In our analysis, this negative effect of agriculture was particularly evident in the Western Siberian grain belt in Russia (Extended Data Fig. 5). Here, the strength of edge effects was comparable to those seen in tropical forests but was driven primarily by the high fraction of agriculture rather than climatic factors such as MAT and MAP, which are more influential in the tropics.

 

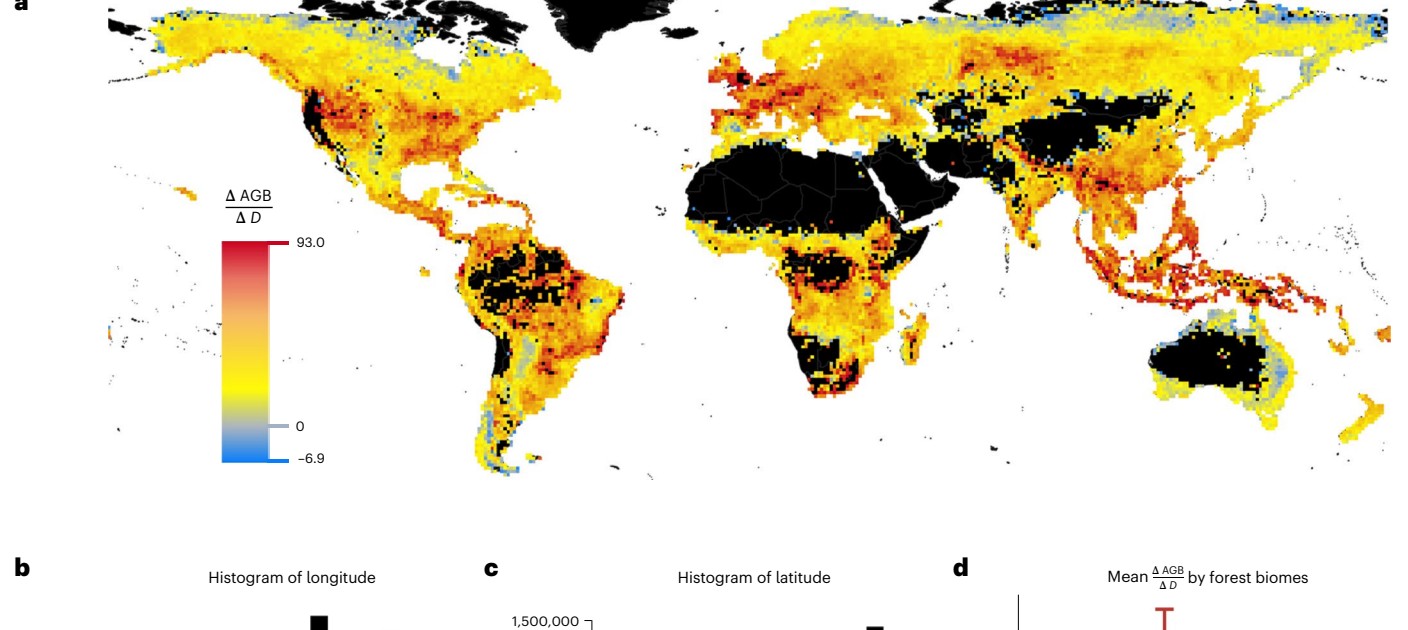

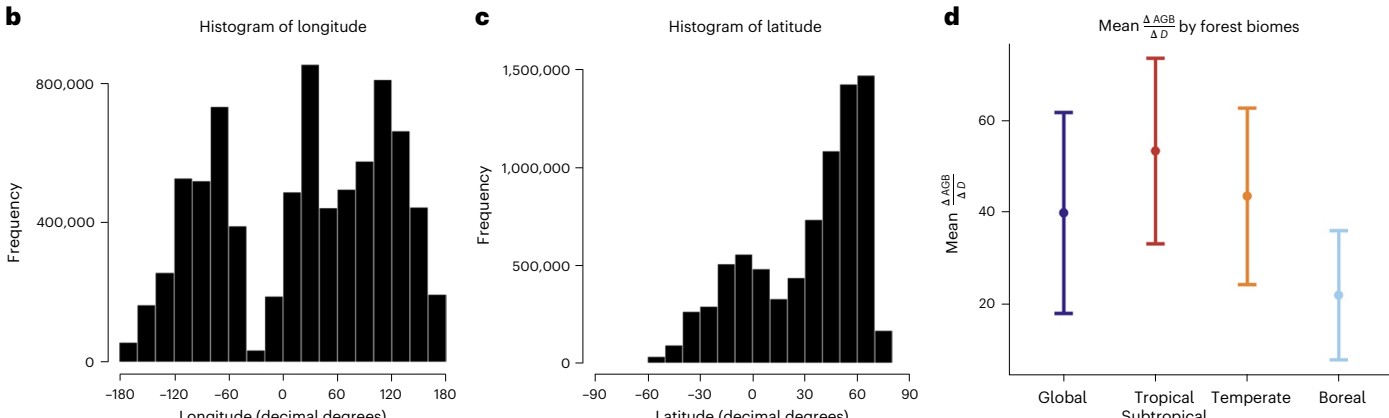

**Fig. 1 | Direction and magnitude of edge effects on forest biomass density.** **a**, Map of edge effects ($\frac{\Delta \text{Aboveground biomass density}}{\Delta \log_{10}(\text{Distance to edge})}$ or $\frac{\Delta \text{AGB}}{\Delta D}$), calculated individually within each 100-km grid cell using spatial log-linear regression (Methods). A more positive $\frac{\Delta \text{AGB}}{\Delta D}$ corresponds to a greater decrease in aboveground biomass density near the forest edge relative to the interior. **b,c**, Histograms of longitude (**b**) and latitude (**c**) for sampled forest locations. **d**, Mean ± s.d. of $\frac{\Delta \text{AGB}}{\Delta D}$ by forest biomes weighted by inverse coefficients of variation.

The third most important variable in the global analysis, MAP, emerged as the top predictor of edge-effect magnitude within tropical biomes (Fig. 2a,f). Near forest edges, increased air and soil temperatures and increased wind exposure raise VPD, heightening vegetation vulnerability to water stress[38]. While drought-adapted plants in tropical dry forests possess biochemical and morphological mechanisms to conserve water[39], forests in water-limited but not drought-adapted areas appear particularly susceptible. This is reflected in the SHAP dependence plot, where SHAP values increase with MAP up to a threshold, indicating stronger negative edge effects in regions with moderate precipitation (Fig. 2d). In regions with extremely high precipitation (>3,000 mm), the impact of MAP on edge effects diminished (Fig. 2d), probably because water is not limiting in these areas. This finding aligns with previous research, which indicates that water availability primarily limits plant growth in ecosystems receiving less than ~2,000 mm of precipitation[6,40].

Several interactions were evident among the key predictors (Fig. 2d,e). First, at higher MAT, SHAP values of MAP increased more sharply with rising MAP compared with lower MAT (Fig. 2d). This suggests a synergistic effect between high temperature and precipitation in amplifying edge effects. One likely explanation is that elevated MAT increases VPD near forest edges, intensifying moisture gradients between edge and interior environments, particularly in forests that are

not drought adapted[38]. Second, interactions between agriculture and MAP were particularly pronounced in drier regions (MAP < 1,000 mm; Fig. 2e). In such areas, agricultural activity can substantially alter atmospheric and soil moisture dynamics[41,42], exacerbating local water stress. Widespread use of irrigation in low-precipitation regions[43] may create artificially wetter conditions at forest edges, reducing the microclimatic contrast with wetter forest interiors. This could help explain the weaker edge effects (reflected by steeper SHAP value declines) in regions with low MAP.

To further examine the influence of environmental variables, we ran an XGBoost model using edge effects on tree canopy cover[30] ($\frac{\Delta \text{Tree cover}}{\Delta D}$, Extended Data Fig. 3) as the response variable instead of biomass-based edge effects ($\frac{\Delta \text{AGB}}{\Delta D}$). Overall, the results were broadly consistent; however, notable differences emerged in regions with high agricultural land cover (>75%), where forest cover is minimal, and in tropical regions (Extended Data Fig. 6). These discrepancies probably stem from the fact that tree cover does not directly equal biomass density, as biomass is additionally influenced by tree height, diameter and wood density. Notably, in tropical regions, the edge effect on tree cover was less negative than that on biomass, suggesting that tree cover near edges remained relatively stable despite declines in biomass. Indeed, large trees at the tropical edge tend to be thinner and shorter, yet retain

similar crown width compared to interior trees[44]. These edge-induced changes in tree architecture may explain how canopy cover remains relatively stable while biomass declines, especially in structurally complex forests such as those of the Amazon[44].

The observed contributions of macro-scale environmental covariates to global edge-effect variation, identified by our XGBoost and SHAP analysis, are probably modulated by fine-scale factors such as soil conditions and forest microclimate. Changes in soil conditions near forest edges, including reduced soil carbon, lower enzyme activity and altered soil texture with increased freeze–thaw cycles[11], can limit tree growth and biomass accumulation by elevating nutrient constraints and environmental stress. Similarly, the observed reduction in canopy cover at edges (Extended Data Fig. 3) weakens forests' microclimatic buffering capacity, leading to greater temperature fluctuations and local heating[45,46]. This in turn could amplify the sensitivity of edge environments to macroclimatic factors such as temperature and precipitation[46]. Future studies incorporating fine-scale soil and microclimatic data will be needed to explicitly test the mechanistic roles of each variable and disentangle the interactions between these local factors and the broader environmental drivers that explain global variation in edge effects.

## AGB difference between edge and interior

To assess the impact of edge effects on forest carbon stocks, we quantified how biomass differences between forest edges and interiors scale up to influence total forest AGB. Within each grid cell, we compared the observed AGB in edge areas to the expected AGB, as predicted by our spatial log-linear regression models at the grid-cell level. To construct a counterfactual scenario without edge effects, we assumed that edge areas would have the same biomass density as nearby forest interiors. Edge areas were defined by the depth of edge influence, a threshold distance beyond which edge effects are considered to dissipate (Fig. 3a; see Methods). The global mean depth of edge influence was 336 m, with biome-specific averages of 826 m for tropical forests, 235 m for temperate forests and 258 m for boreal forests. Globally, mean biomass density in edge areas was 16% lower than in interior forests. Using this depth-of-influence framework, we quantified the 'missing biomass' (sensu ref. 19) as the difference between observed edge-affected AGB and the counterfactual AGB expected in the absence of edge effects.

The spatial distribution of missing biomass followed the global patterns of edge effects, with tropical biomes showing the highest AGB losses due to their pronounced edge effects (Figs. 3b and 1d). Globally, we estimated a cumulative AGB loss of 58 Pg (95% confidence interval (CI): 49–68 Pg) (Fig. 3c), representing a 9% AGB decrease relative to a counterfactual scenario without edge effects. At the biome level, estimated AGB losses amounted to 28 Pg (95% CI: 23–32 Pg; 7% of the biome's total biomass) in tropical/subtropical forests, 11 Pg (95% CI: 9–12 Pg; 10%) in temperate forests and 8 Pg (95% CI: 6–9 Pg; 11%) in boreal forests (Fig. 3c).

This global biomass loss of 58 Pg is more than twice the AGB of all forests in Europe excluding the Russian Federation[5]. Translating biomass to carbon, this equals 28 Pg C of aboveground carbon loss, assuming a mean wood carbon concentration of 47.6% (ref. 47). When accounting for belowground carbon stored in roots (assumed to comprise 22% of total tree biomass[48]), total carbon losses rise to ~36 Pg C.

These results highlight the substantial contribution of forest fragmentation to biomass loss and underscore the need for incorporating edge effects into carbon stock assessments using standardized methodologies. Our analysis addresses this using the biomass data from the year 2000[31], as this dataset offers the highest-resolution (30 m) global biomass map currently available. However, substantial forest changes have occurred since 2000. For example, temperate forests in East Asia and parts of the boreal zone have experienced tree cover gains, while tropical regions such as the Amazon and Southeast Asia have seen substantial losses[49]. These changes could potentially impact our results if they were accompanied by disproportionate biomass gains or losses between forest edges and interiors. For example, deforestation often exposes interior forests to edge conditions, leading to faster biomass loss, while regrowth near edges can be slower due to harsher environments. Together, these dynamics can amplify overall biomass loss across the landscape. In heavily fragmented landscapes, true interior forests are becoming scarce, complicating the detection of edge–interior biomass gradients. Despite this limitation, our analysis provides a robust baseline for understanding global edge effects around the year 2000. Future studies could build on this work by assessing the sensitivity of edge effect estimates to temporal changes, using newer biomass datasets where available, even if at coarser spatial resolutions or for specific regions.

## Importance of context dependency

Previous studies in temperate forests have suggested higher biomass and tree basal area near forest edges[20,50,51], often attributed to factors such as elevated nitrogen deposition from anthropogenic sources and increased light availability near edges[50]. These studies were typically conducted at the field scale and focused on the first 30 m from the forest edge, a distance that would fall entirely within a single pixel in our 30-m-resolution global analysis. As a result, such fine-scale edge enhancements may not be detectable in our study. These contrasting patterns highlight the importance of considering both spatial scale and methodology when interpreting edge effects. While localized field studies provide valuable mechanistic insights at fine spatial resolution, our approach is designed to detect broader, landscape-scale patterns of biomass variation that extend well beyond the immediate forest edge.

The large-scale findings from the United States by ref. 20 using national forest inventory data (FIA plots) suggest higher tree density (number of trees per ha) and basal area in forest edges compared with interior forests. However, their analysis defined forests as areas with ≥10% canopy cover[20], whereas we applied a more conservative threshold of 30%. As a result, low canopy cover areas that were classified as 'interior' in their analysis would have been excluded entirely from our study. This difference may have contributed to their relatively lower estimates of interior forest biomass. Moreover, while ref. 20 used a binary classification to distinguish 'edge' from 'interior' plots, our approach modelled biomass as a continuous function of distance to edge.

To better understand the differences between our findings and those of ref. 20, we conducted an additional analysis using plot-level FIA data[52] for the United States from the year 2000. Our objective was to replicate their binary 'edge' versus 'interior' classification on the basis of proximity to forest edges (see Methods) and assess whether

**Fig. 2 | Contribution of environmental variables to edge effects ($\frac{\Delta AGB}{\Delta D}$).**
**a**, SHAP summary plot showing the contribution of each environmental variable to predicted edge effects across grid cells. Variables are ranked by their mean absolute SHAP value (|SHAP|), with the most influential variables listed at the top. The $x$ axis indicates the SHAP value (that is, contribution to prediction) and each dot represents a local (grid-cell level) SHAP value. The overall distribution of points illustrates the global importance of each variable. **b,c**, SHAP dependence plots for MAT (**b**) and agricultural land cover (**c**). The $x$ axis shows the variable value and the $y$ axis shows its corresponding SHAP value. Colour shading reflects the density of data points, with lighter colours indicating higher density. **d,e**, SHAP dependence plot for MAP, coloured by MAT (**d**) and agriculture (**e**) to illustrate interaction effects between variables. The red lines in **b**–**e** represent LOESS-smoothed trends. **f**, Dominant environmental variable by biome. Each grid cell is coloured according to the SHAP value of the single most important variable, defined by the highest mean |SHAP|, from biome-specific XGBoost models for tropical/subtropical, temperate and boreal forests. Only the top variable per biome is shown; for full variable sets and biome-specific results, see Methods and Extended Data Fig. 4.

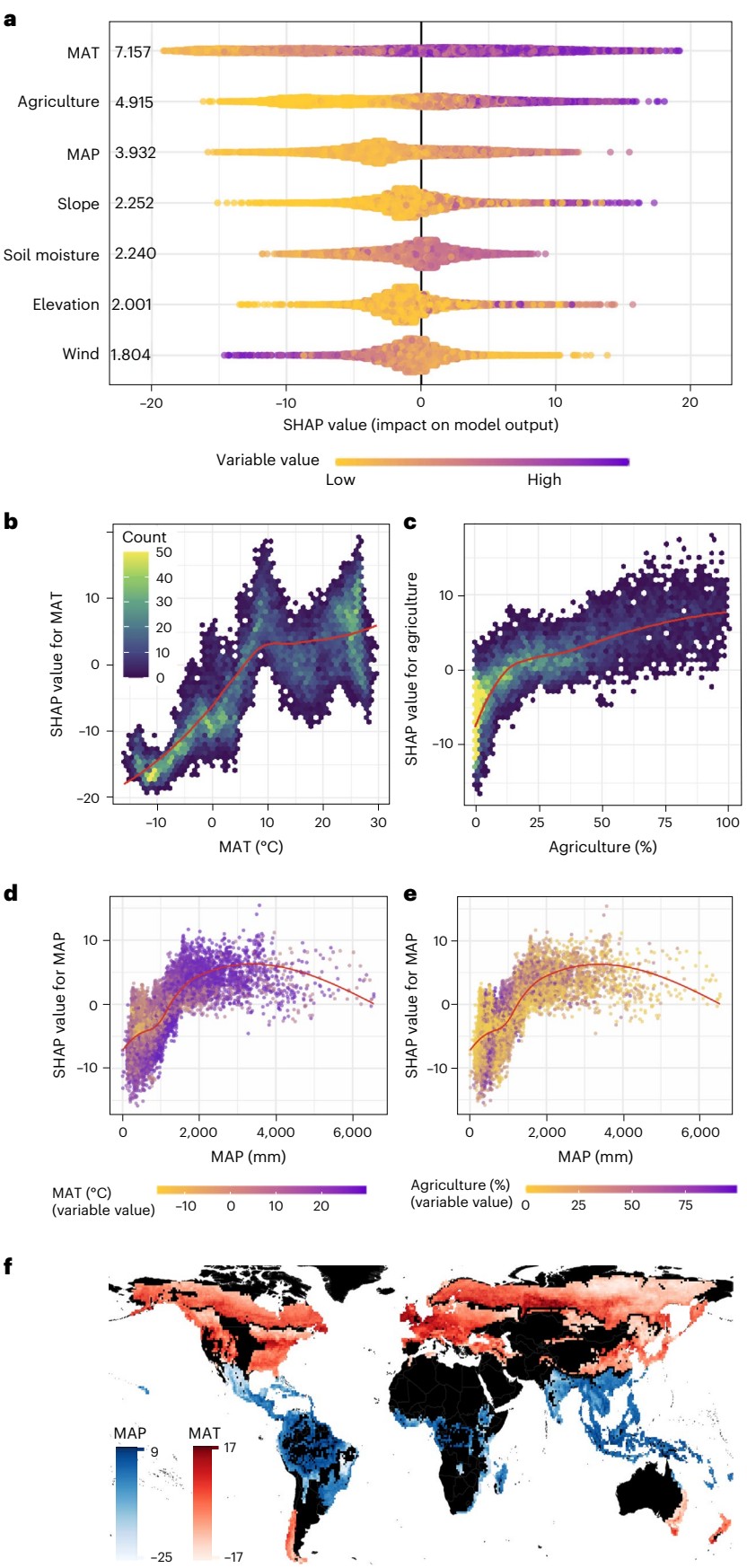

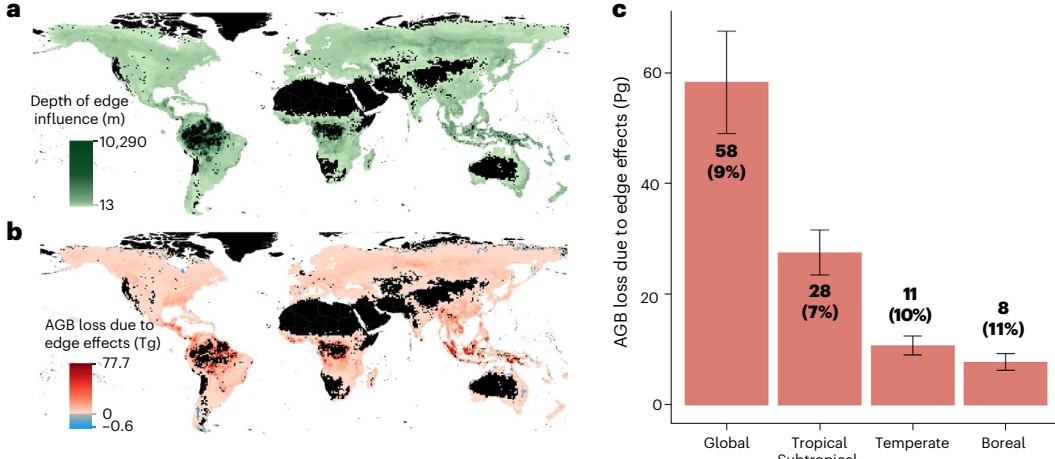

**Fig. 3 | Quantification of AGB considering forest edges. a**, Estimated depth of edge influence (in meters) at the grid-cell level, defined as the threshold distance beyond which biomass density stabilizes and no longer exhibits a notable gradient with respect to edge proximity. **b**, Spatial distribution of AGB loss (in teragrams, Tg) attributable to edge effects at the grid-cell level. AGB loss is calculated as the difference between observed biomass and the counterfactual biomass if edge areas had the same biomass density as nearby forest interiors. **c**, Total AGB loss due to edge effects, aggregated at global and biome levels (in petagrams, Pg). The numbers in parentheses indicate the percent reduction in observed AGB relative to a counterfactual scenario without edge effects. Error bars represent 95% CIs.

applying a similar approach would yield comparable results. Our analysis also included an 'intermediate' category for a more detailed comparison. Basal area[52], tree density[52] and biomass density values[53] were analysed to compare these categories. Our analysis showed that all three metrics were significantly higher in interior plots compared with edge plots (Kruskal–Wallis and Dunn's test, $P < 0.001$; Extended Data Fig. 7). Intermediate plots showed values similar to interior plots, especially for biomass density and basal area (Kruskal–Wallis and Dunn's test, $P > 0.05$). These results were robust across different threshold definitions for edge and interior classification (Extended Data Fig. 7). This plot-level analysis corroborates our findings and highlights the critical role of methodological definitions of forest in interpreting edge effects. The results suggest that edge effects on forest biomass are context dependent, varying with landscape fragmentation and forest canopy cover. Further research is needed to explore how definitions of forest and edge influence outcomes, particularly in human-impacted areas such as agricultural land with low tree cover (10–30%).

## Conclusions

High temperature emerged as a key predictor of negative edge effects (Fig. 2), highlighting the potential for climate warming to amplify edge-induced forest degradation and carbon loss. Our results suggest that regional climate warming and localized temperature increases at forest edges[9] may interact synergistically to drive reductions in aboveground biomass. This effect was particularly pronounced in tropical and temperate forests, where combined warming pushes trees beyond physiological limits, increasing heat stress, reducing growth and elevating mortality rates[54]. While absolute biomass density differences between edges and interiors were smaller in boreal forests due to their overall lower biomass, edge effects led to substantial relative aboveground biomass loss of 11% (Fig. 3c). Given that boreal forests are anticipated to experience disproportionately severe warming compared with other biomes[1], these findings highlight their heightened vulnerability. Rapid warming may exceed the adaptive capacities of boreal forest vegetation, exacerbating susceptibility to disturbances such as wildfires and drought[55,56]—disturbances already intensified at forest edges due to their exposed and drier conditions.

The widespread prevalence of negative edge effects (Fig. 2a) suggests a troubling synergy between two potent global change pressures: forest fragmentation and climate warming. Fragmentation exposes more forested areas to edge effects, while rising temperatures exacerbate biomass loss at these edges. Together, these forces pose a serious threat to the global forest carbon sink, with potentially compounding negative feedbacks to climate. While direct forest removal is responsible for 15–25% of annual human carbon emissions[31], our work reveals the substantial indirect carbon costs of forest fragmentation. Our findings call for an urgent reorientation of forest management and conservation strategies to address the dual threats of deforestation and fragmentation. Policies must not only prioritize reducing forest loss but also mitigate the indirect impacts of fragmentation to preserve the critical role of forests as a global carbon sink.

## Methods

### Data acquisition and preparation

We used the 30-m-resolution forest cover map for the year 2000 published by ref. 30. We defined forest as a pixel where 30% or greater is covered by trees taller than 5 m in height. To compare variations in edge effects across the globe, we used Google Earth Engine[57] to cover the Earth's land surface with a grid where each cell is 100 km × 100 km in size. Then, we sampled 500 random points per grid cell and calculated, for each point falling within a forest, the Euclidean distance in meters between it and the nearest non-forest pixel. For each point, we also measured aboveground forest biomass density using the 30-m-resolution map of aboveground forest biomass for the year 2000 produced by ref. 31. The ref. 31 2000 map represents the most comprehensive global aboveground biomass dataset available at this fine 30-m resolution. To ensure temporal consistency, we opted for the 2000 datasets from refs. 30,31, allowing us to directly compare forest cover and biomass at the same historical point.

After filtering out points falling outside of the forest, we retained 8,077,835 biomass/edge measurements across 17,309 grid cells. We retained only grid cells containing more than 20 points to exclude small samples, which may give biased results, resulting in 8,074,224 points. To guarantee the detectability of edge effects, we also excluded grid cells where less than 3% of points fell within 100 m of a non-forest pixel, resulting in a final dataset of 7,837,233 points. Downstream analyses were carried out using the R statistical programming language (v.4.2.1)[58].

## Spatial log-linear regression

Within each individual grid cell, we conducted a spatial log-linear regression:

$$Y = \beta_0 + \beta_1 \times X + \varepsilon \qquad (1)$$

where $Y$ is the aboveground biomass density (Mg ha$^{-1}$), $X$ is the log$_{10}$-transformed distance to the nearest forest edge (m), $\beta_0$ is the intercept, $\beta_1$ is the slope coefficient describing the relationship between biomass density and distance to the edge and $\varepsilon$ is the error term. This model structure was selected to reflect the expected non-linear nature of edge influence, where biomass loss is strongest near the edge and diminishes progressively with distance. Since we already excluded grid cells with less than 20 points, no additional statistical methods were used to predetermine sample size. To ensure that spatial autocorrelation did not bias our results, we used spatially filtered log-linear regression with eigenvector-based spatial filtering as implemented in R package spfilteR[59].

Estimate values of $\beta_1$, which quantify the observed local relationship between biomass density and distance from the forest edge within each individual grid cell, were extracted. Outliers of extreme 2.5% values, corresponding to 377 values from each side, were discarded, resulting in 15,094 $\beta_1$ values. We refer to $\beta_1$ as $\frac{\Delta \text{AGB}}{\Delta D}$, which is an abbreviated form of $\frac{\Delta \text{Aboveground biomass density}}{\Delta \log_{10}(\text{Distance to edge})}$. Positive $\frac{\Delta \text{AGB}}{\Delta D}$ indicates a positive relationship between biomass density and log$_{10}$-transformed distance variables; in other words, a relationship where biomass density is lower near the edge. A negative $\frac{\Delta \text{AGB}}{\Delta D}$, on the other hand, indicates that biomass density is higher near the edges. We measured the mean and standard deviation of $\frac{\Delta \text{AGB}}{\Delta D}$ at a global scale as well as tropical/subtropical, temperate and boreal biome scales, weighted by corresponding inverse coefficients of variation[60]. To ensure that inaccuracies in the aboveground forest biomass map do not cause us to erroneously detect edge effects, we also performed the spatial log-linear regression using tree canopy cover[30] as a response variable instead. We found analogous results to our primary analysis (Extended Data Fig. 3), demonstrating that lowered forest biomass near edges is most likely attributable to decreases in canopy cover rather than any kind of data product error.

In addition, we analysed edge effects using forest inventory data[52] for the United States using FIA plots from 2000. Because FIA plot locations have a built-in positional uncertainty, with coordinates randomly displaced within a 1-mile (1,609.34 m) radius for privacy reasons, we implemented a buffer-based approach to account for this imprecision. We created a 1,609.34 m buffer zone around each plot and calculated the average distance to the nearest forest edge within this buffer. Distance calculation was done using the forest cover map from ref. 30, as in the main analysis. We then classified plots into three categories: 'interior' (plots located more than 150 m from the nearest edge), 'edge' (plots within 50 m of the nearest edge) and 'intermediate' (plots between 50 and 150 m from the edge). For each plot, basal area and tree density values were directly derived from the FIA data, while biomass density values for each plot were obtained from ref. 53. To identify possible differences in these metrics among edge, intermediate and interior plots, we applied Kruskal–Wallis and Dunn's tests. To ensure the robustness of our classifications, we varied the edge and interior thresholds by increasing the edge threshold from 50 m to 100 m and the interior threshold from 150 m to 200 m. We then reran Kruskal–Wallis and Dunn's tests to assess the sensitivity of our findings to these threshold changes (Extended Data Fig. 7).

In our analysis, we seek to examine the influence of large-scale environmental gradients on edge effects rather than to explain or statistically account for variation in edge effects at small scales within individual grid cells. For this reason, we include environmental covariates only in our downstream XGBoost models, rather than in these grid-cell level spatial regressions. The grid-cell level spatial regressions are intended to simply measure the actual, realized amount of edge effect occurring in each grid cell, without reference to any potential environmental drivers (which are addressed downstream). This approach allows us to meaningfully examine variation at larger scales.

## Machine-learning models and interpretation

To examine how environmental gradients influence the strength of edge effects across the globe, we used an interpretable machine-learning approach to identify the drivers of variation in the $\frac{\Delta \text{AGB}}{\Delta D}$ estimates from our grid-cell level spatial log-linear regressions. To do this, we first selected environmental covariates on the basis of previous literature, verifying that no covariates were highly correlated with one another (Spearman's rank correlation coefficients <0.7). These environmental covariates included mean annual temperature (MAT), mean annual precipitation (MAP), mean annual wind speed, mean soil moisture, elevation, slope, and percentage of cultivated and managed vegetation (referred to as agriculture) (see Extended Data Table 2 for details of environmental covariates). We calculated their mean values within each grid cell using Google Earth Engine[57].

Then, we fit an XGBoost model[32] to investigate how environmental factors influence the direction and magnitude of the edge effects ($\frac{\Delta \text{AGB}}{\Delta D}$) across the globe[33]. Since we had already directly measured grid-cell level edge effects in our upstream analysis, our goal with the XGBoost model was not to perform prediction but to empirically identify primary drivers of variation. With 80% of the original data used as training data, hyperparameters were tuned by Bayesian optimization[61]. To propagate uncertainty in the edge-effects estimates from our spatial log-linear regressions, each training data point was weighted by its inverse coefficient of variation (see Extended Data Fig. 1 for the map of coefficients of variation). The XGBoost model performance was measured by calculating root mean squared error (RMSE), $R^2$ and mean absolute error (MAE) metrics on the test dataset corresponding to the randomly selected 20% of the original data. The final XGBoost model showed RMSE = 13.02, $R^2$ = 0.67 and MAE = 9.44. We also measured these metrics using the spatial leave-one-out method with various spatial buffer sizes (Extended Data Table 1).

We also fit biome-level XGBoost models for tropical/subtropical, temperate and boreal forests separately, following the biome designations of ref. 62. Grid cells that straddled multiple biomes were excluded from these models. As before, highly correlated (Spearman's rank correlation coefficients >0.7) and less important environmental covariates according to the global-scale model were also excluded (see Extended Data Fig. 4 for the list of environmental covariates used in each biome-scale model). Model performance evaluation was carried out with the same approach as for the global model. The biome-level XGBoost models showed RMSE = 16.97, $R^2$ = 0.40 and MAE = 13.34 for tropical forests, RMSE = 14.71, $R^2$ = 0.50 and MAE = 11.27 for temperate forests, and RMSE = 6.79, $R^2$ = 0.70 and MAE = 5.08 for boreal forests. In addition, we fit an XGBoost model using edge effects on tree canopy cover[30] (that is $\frac{\Delta \text{Tree cover}}{\Delta D}$ in Extended Data Fig. 3) as the response variable, which yielded RMSE = 5.30, $R^2$ = 0.64 and MAE = 3.92.

Finally, we performed SHAP analyses to interpret the XGBoost models[63]. SHAP analysis is based on additive feature attribution methods, where a particular prediction is explained as a sum of the contribution values of individual input features[64]. SHAP values represent these contribution values. For our XGBoost model, the sum of the SHAP values of all environmental variables results in the estimated $\frac{\Delta \text{AGB}}{\Delta D}$ for the corresponding grid cell. A high positive and high negative SHAP value of a variable contributes to predicting the high positive and high negative $\frac{\Delta \text{AGB}}{\Delta D}$, respectively. The |SHAP| value represents the degree of contribution of the variable to the local prediction. We obtained mean |SHAP| values to compare the global contribution magnitude of each variable[32]. We visualized model interpretation with the SHAP summary plot and SHAP dependence plots.

## Quantification of missing biomass

To estimate the impact of edge effects on the global carbon cycle, we quantified AGB at a global and forest-biome scale. We used the extent of our forest pixels to measure the global scale AGB. For the forest-biome-scale AGB, we delineated the extent of the tropical/subtropical, temperate and boreal forest biomes following ref. 62. We measured the 'missing biomass' sensu ref. 19 by comparing the observed AGB produced by ref. 31 with a counterfactual 'expected AGB', which is the AGB that would be expected if forest edge areas showed the same biomass density as nearby forest interiors. To this end, we used our grid-cell-level spatial log-linear regression models (equation 1).

Specifically, we first defined a distance threshold separating 'edge' areas from 'interior' areas. Since we found that the strength of edge effects varies globally, we reasoned that it was important to vary this distance threshold according to our observations rather than to impose a single, uniform threshold value across all regions. Thus, the distance threshold was defined within each grid cell as the mean distance to the edge for points with the 90th percentile of biomass density for that grid cell. This logic is similar to that of ref. 19, which defined the depth at which 90% of asymptotic biomass is reached. We refer to this threshold distance as the 'depth of edge influence'. We classified forest pixels that were closer to the edge than the depth of edge influence as edge areas and those at or beyond the depth of edge influence as interior areas.

Then, the biomass density of each forest pixel (in Mg ha$^{-1}$) was multiplied by the pixel area (in ha) to calculate the pixel's AGB stock (in Mg). The actual AGB was calculated as the sum of the AGB of all pixels within each forest biome. To calculate the counterfactual expected AGB without edge effects, the AGB values for all pixels within the depth of edge influence were replaced with the value predicted for the depth of edge influence, using the relevant spatial regression model intercepts ($\beta_0$) and $\frac{\Delta AGB}{\Delta D}$ ($\beta_1$) estimated for each grid cell. For example, if the depth of edge influence was equal to 200 m, the AGB of all forest pixels within 200 m from the nearest edge would be replaced with $\beta_0 + \beta_1 \times \log_{10} 200$ according to equation (1). The expected counterfactual AGB would then be calculated as the sum of the interior AGB and the newly estimated edge AGB.

Finally, we calculated the absolute difference of AGB (counterfactual AGB–actual AGB, in Pg) and the percentage difference of AGB ($\frac{absolute\ difference\ of\ AGB}{counterfactual\ AGB} \times 100$, in %). To estimate the uncertainty in AGB differences, we first calculated the 95% CI for each $\frac{\Delta AGB}{\Delta D}$ using standard error-based bounds (that is, lower CI = (Estimate value of $\frac{\Delta AGB}{\Delta D}$) − 1.96 × s.e.; upper CI = Estimate value of $\frac{\Delta AGB}{\Delta D}$ + 1.96 × s.e.). We then applied the same approach to compute the lower and upper confidence intervals for absolute AGB differences, ensuring consistency in uncertainty propagation. We explored different definitions of depth of edge influence, which showed similar patterns across biomes but naturally varied in the total amount of estimated missing biomass (see Extended Data Fig. 8 for an example).

### Reporting summary

Further information on research design is available in the Nature Portfolio Reporting Summary linked to this article.

## Data availability

The dataset of eight million forested locations, including distance to forest edge and corresponding aboveground biomass density, is publicly available in figshare at https://doi.org/10.6084/m9.figshare.29425154 (ref. 65) under a Creative Commons licence (CC BY 4.0). The distance to forest edge data are derived from ref. 30, and the aboveground forest biomass data are sourced from ref. 31. Map-based figures in this study (Figs. 1a, 2f and 3a,b, and Extended Data Figs. 1–3, 5 and 8a,b) include basemaps from the Comprehensive Global Administrative Zones (CGAZ) dataset provided by the geoBoundaries database[66].

## Code availability

The code used for this study is available in figshare at https://doi.org/10.6084/m9.figshare.29425154 (ref. 65) for use under a Creative Commons licence (CC BY 4.0).

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

## Acknowledgements

G.Y., T.W.C., T.L., C.M.Z. and G.R.S. acknowledge funding from ETH Zürich, DOB Ecology and the Bernina Initiative. G.Y. was also funded by the French Foreign Ministry (102390W). G.R.S. was also funded by Ambizione Grant PZ00P3_216194 from the Swiss National Science Foundation.

## Author contributions

G.Y. and G.R.S. conceived the ideas and designed the methodology, refined in discussion with T.W.C. and C.M.Z. T.L. generated the dataset. G.Y. led the analysis. All authors contributed to the interpretation of the results. G.Y. wrote the paper with input from all other authors.

## Funding

## Competing interests

The authors declare no competing interests.

## Additional information

**Extended data** is available for this paper at https://doi.org/10.1038/s41559-025-02840-2.

**Correspondence and requests for materials** should be addressed to Gayoung Yang.

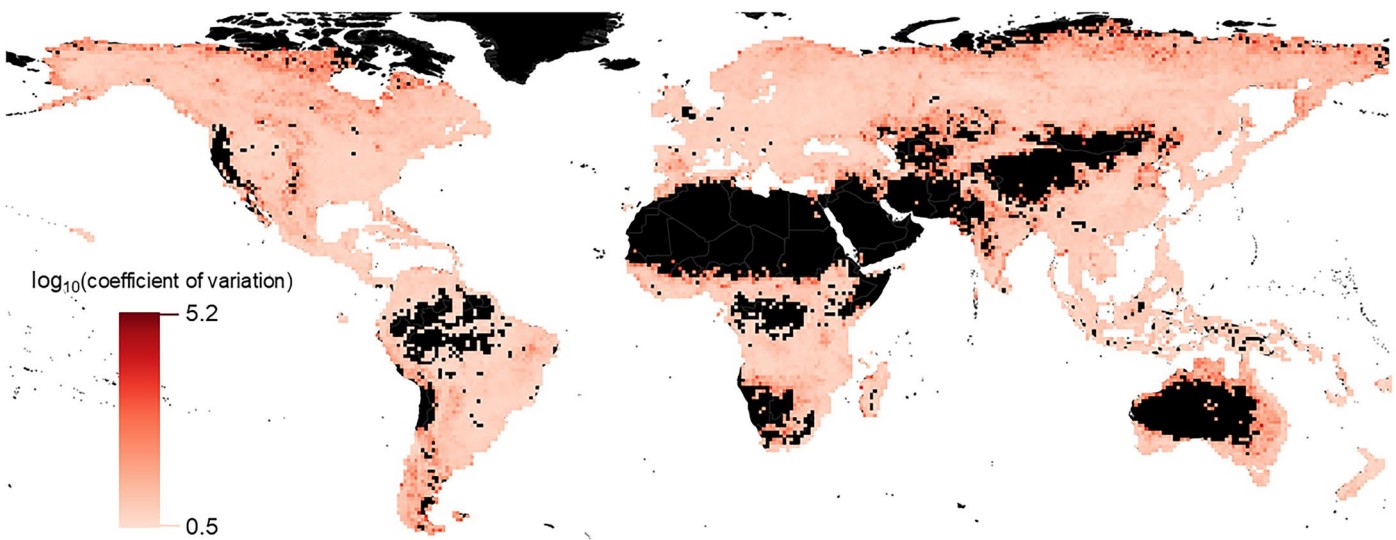

**Extended Data Fig. 1 | Map of the log$_{10}$-transformed coefficients of variation of $\frac{\Delta AGB}{\Delta D}$.** Each geographical grid cell with a scale of 100 km is shaded according to the log$_{10}$-transformed coefficient of variation.

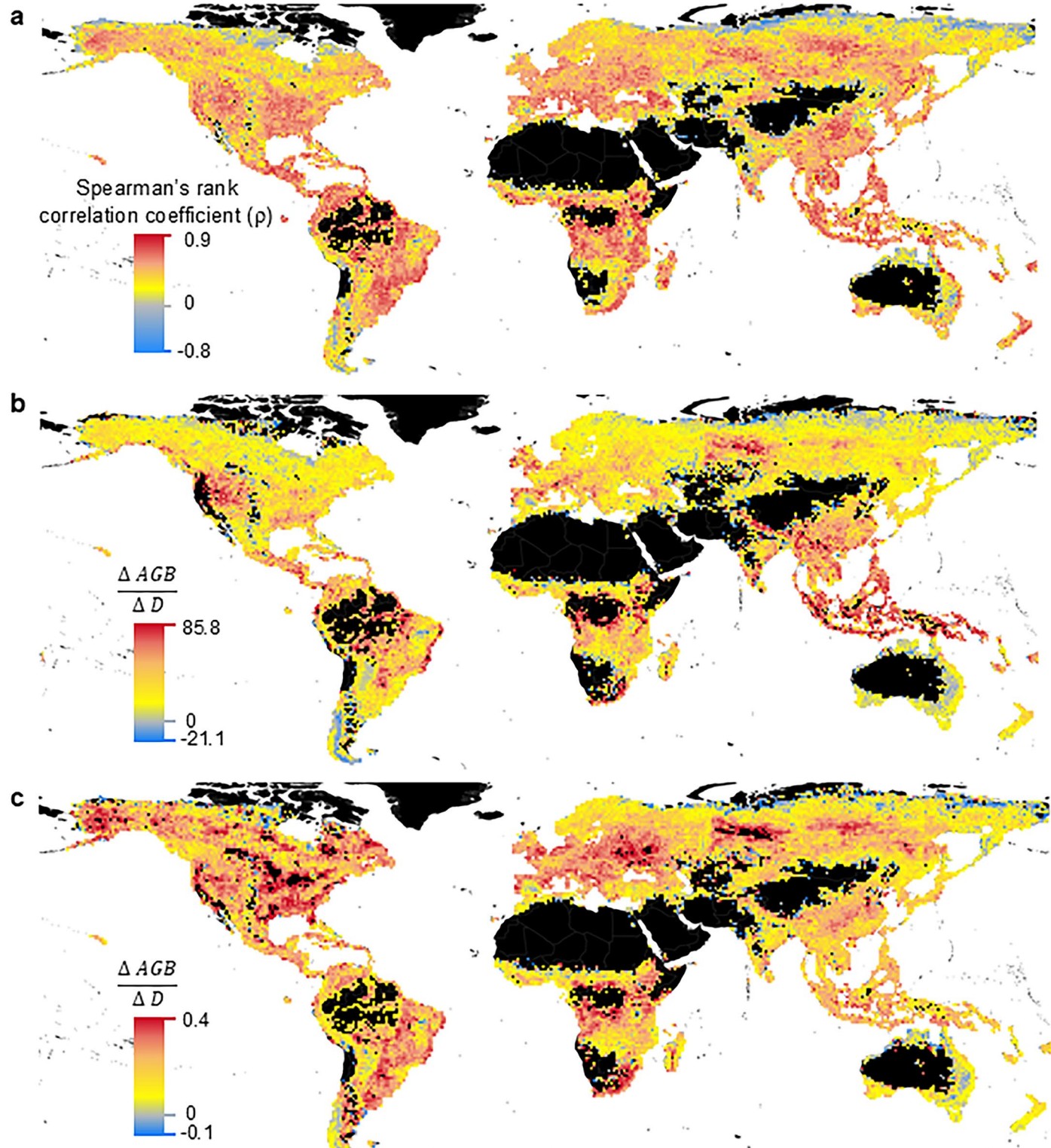

**Extended Data Fig. 2 | Alternative measures of edge effects. a**, Map of the Spearman's rank correlation coefficient ($\rho$). Spearman correlation tests between $\log_{10}$-transformed distance to the nearest forest edge and forest biomass density were conducted by each geographical grid cell with a scale of 100 km. Grid cells are shaded according to their $\rho$ values. **b**, Map of $\frac{\Delta AGB}{\Delta D}$ after filtering out points at the outermost (distance < 30 m) edge, which could be affected by mixed land cover. Spatial log-linear regression models, as described in Methods, are then applied for each grid cell to calculate $\frac{\Delta AGB}{\Delta D}$. **c**, Map of $\frac{\Delta AGB}{\Delta D}$ on $\log_{10}$-transformed biomass density. A spatial log-linear regression model, as described in the Methods section, was conducted per grid cell with $\log_{10}$-transformed biomass density as a dependent variable and $\log_{10}$-transformed distance to the edge as an independent variable. Each geographical grid cell with a scale of 100 km is shaded according to the $\frac{\Delta AGB}{\Delta D}$.

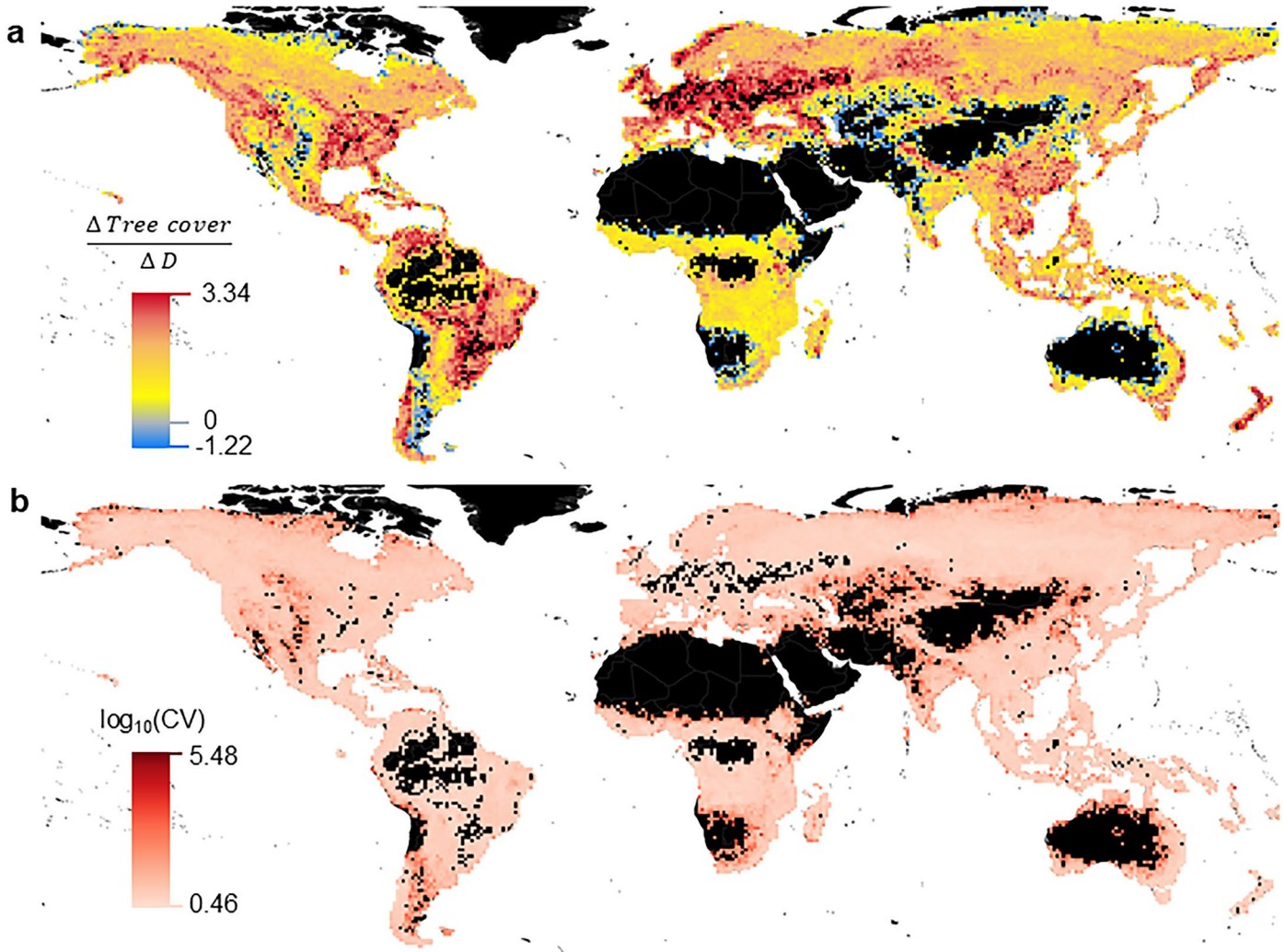

**Extended Data Fig. 3 | Edge effect measurement based on tree cover.** A spatial log-linear regression model, as described in the Methods section, was conducted per grid cell with tree canopy cover for year 2000 from Hansen et al.[30] as a dependent variable and $\log_{10}$-transformed distance to the edge as an independent variable. We refer to estimate value of $\beta_1$ as $\frac{\Delta Tree\ cover}{\Delta D}$, which is an abbreviated form of $\frac{\Delta Tree\ canopy\ cover}{\Delta \log_{10}(Distance\ to\ edge)}$. **a**, Map of $\frac{\Delta Tree\ cover}{\Delta D}$ on tree cover. Each geographical grid cell with a scale of 100 km is shaded according to the $\frac{\Delta Tree\ cover}{\Delta D}$. **b**, Map of the $\log_{10}$-transformed coefficients of variation of $\frac{\Delta Tree\ cover}{\Delta D}$. Each geographical grid cell with a scale of 100 km is shaded according to the $\log_{10}$-transformed coefficient of variation.

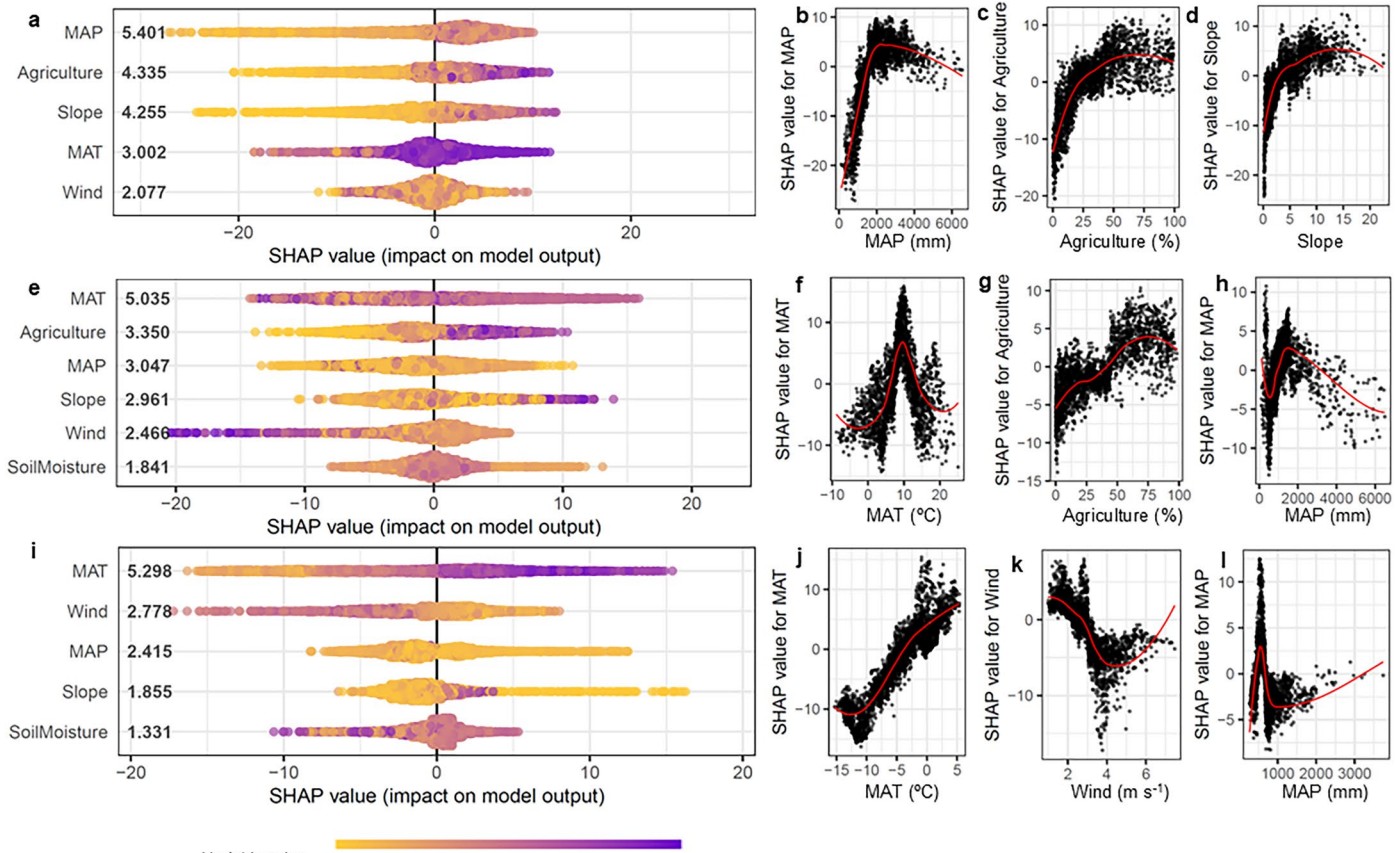

**Extended Data Fig. 4 | SHAP-based interpretation of environmental drivers of edge effects across biomes.** SHAP summary plots and dependence plots showing the contribution of each environmental variable to the estimation of edge effect ($\frac{\Delta AGB}{\Delta D}$) in tropical (**a–d**), temperate (**e–h**), and boreal forests (**i–l**). For each biome, panels show the SHAP summary plot (**a, e, i**) and dependence plots for the top three predictors (**b–d** for tropical, **f–h** for temperate, and **j–l** for boreal). Some variables were excluded due to collinearity: in tropical forests, soil moisture and elevation were excluded due to correlation with MAP and MAT, respectively; in temperate forests, elevation was excluded due to collinearity with slope; and in boreal forests, agriculture and elevation were excluded due to collinearity with MAT and slope, respectively.

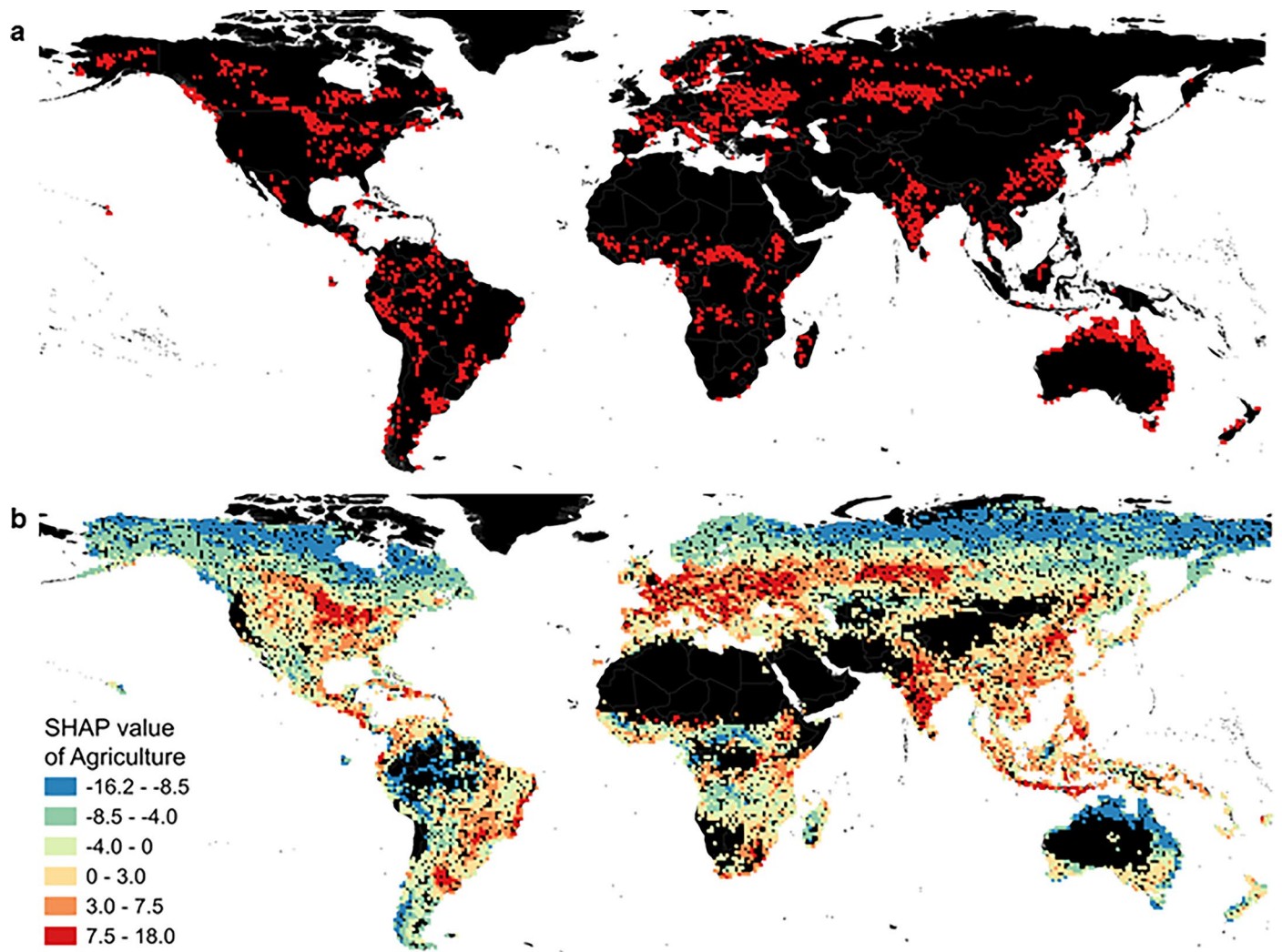

**Extended Data Fig. 5 | Spatially visualized impact of agriculture variable to edge effect estimation. a**, Map of the regions where the agriculture variable contributed the most to the edge effect prediction. **b**, Map of the SHAP value of agriculture.

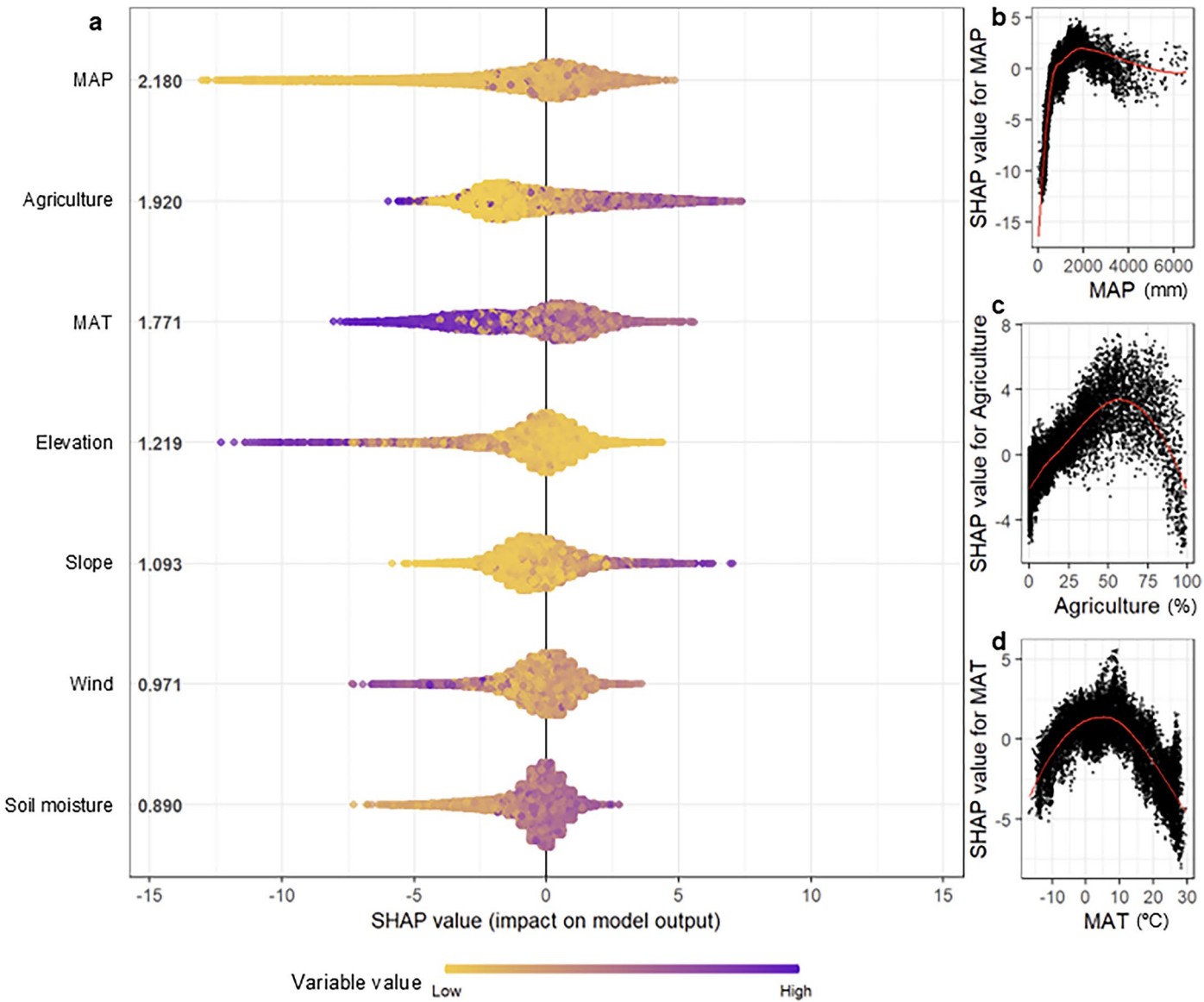

**Extended Data Fig. 6 | Visualization of the contribution of each environmental variable to the edge effects on tree canopy cover ($\frac{\Delta Tree\,cover}{\Delta D}$).** SHAP summary plot (**a**) and dependence plots of top-3 predictors (**b-d**).

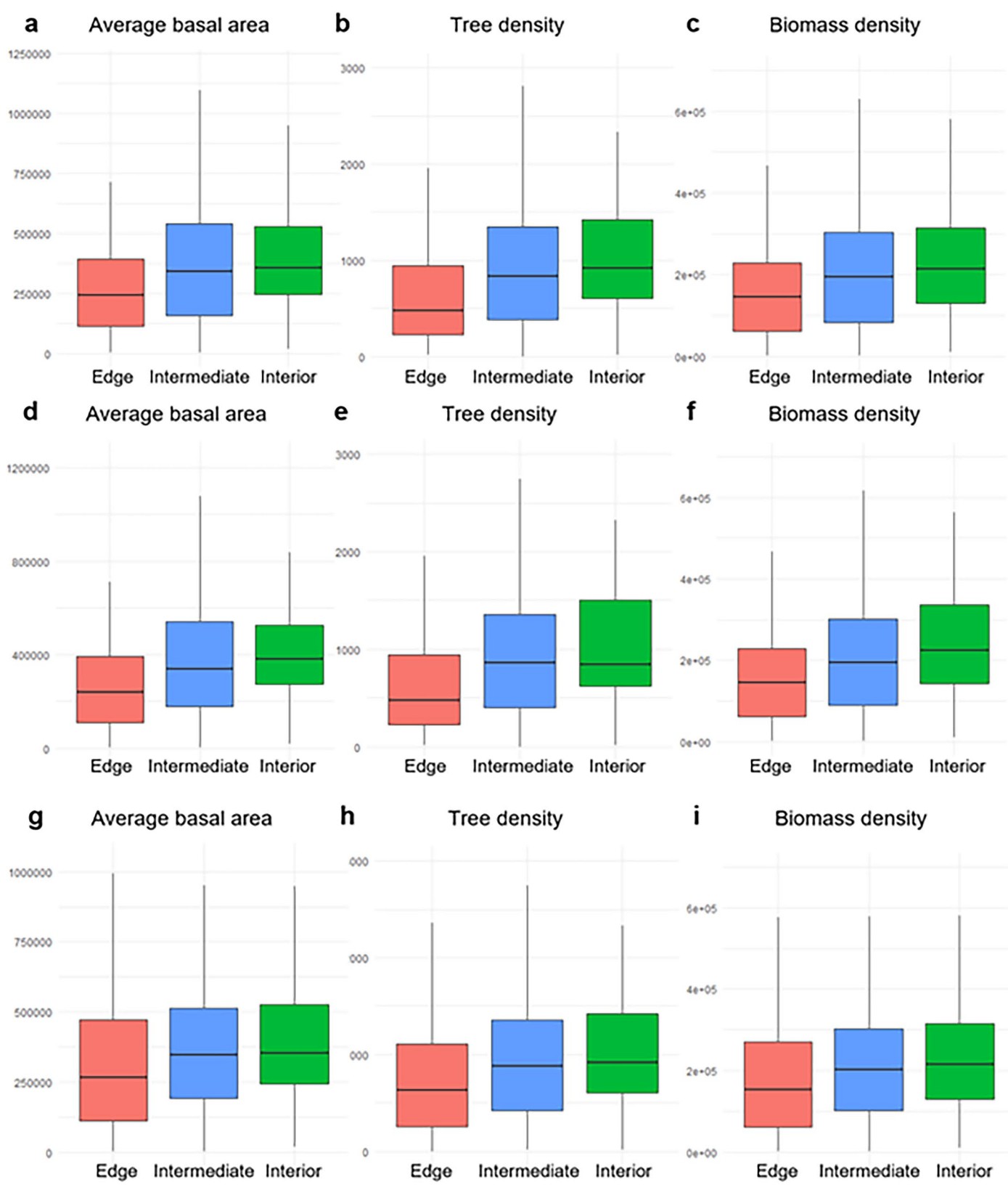

**Extended Data Fig. 7 | See next page for caption.**

**Extended Data Fig. 7 | Comparison of edge, intermediate, and interior plots using data from Global Forest Biodiversity Initiatives (GFBI).** Boxplots illustrate average basal area (cm$^2$) (**a, d, g**), tree density (number of trees per ha) (**b, e, h**), and biomass density (kg ha$^{-1}$) (**c, f, i**). In panels **a-c**, plots with distances less than 50 m, between 50 m and 150 m, and greater than 150 m were classified as edge, intermediate, and interior plots, respectively. These thresholds (50 m and 150 m) correspond to approximately the 25$^{th}$ and 75$^{th}$ percentiles of all distance values. In panels **d-f**, an expanded interior threshold was used, where edge plots are within 50 m, and interior plots start from 200 m. In panels **g-i**, an expanded edge threshold includes edge plots up to 100 m, with interior plots beginning at 150 m. In all panels, boxplots show the median (center line), interquartile range (IQR; box limits), and 1.5×IQR (whiskers). Across all threshold scenarios, average basal area, tree density, and biomass density were consistently higher in interior plots compared to edge plots (Kruskal-Wallis and Dunn's test, $p < 0.001$). Intermediate plots generally showed similar values to interior plots, particularly in average basal area and biomass density (Kruskal-Wallis and Dunn's test, $p > 0.05$). Sample size: n = 679 GFBI plots.

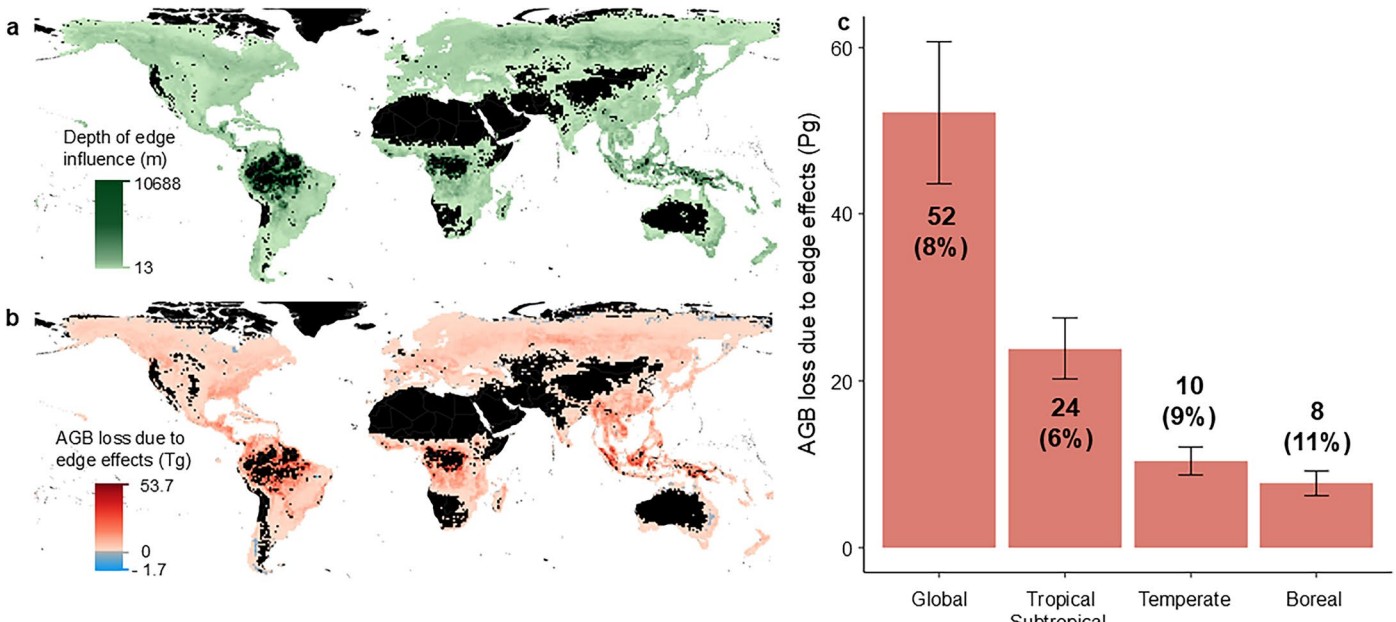

**Extended Data Fig. 8 | Quantification of AGB when depth of edge influence is defined as 75ᵗʰ percentile of the distance to the edge. a**, Depth of edge influence in meters at grid cell level. **b-c**, AGB loss due to edge effects at grid cell level (**b**) and biome level (**c**). AGB loss is defined by the difference between observed and expected AGB assuming that edges show the same biomass density as nearby forest interiors. For (**c**), the percentage in brackets shows how low the observed

AGB is compared to the counterfactual AGB. Mean depth of edge influence values were: 320 m (global), 767 m (tropical/subtropical), 235 m (temperate), and 262 m (boreal). Total AGB loss was: 52 Pg (95% CI: 44–61 Pg; 8% of total biomass) globally, 24 Pg (95% CI: 20–28 Pg; 6%) for tropical/subtropical, 10 Pg (95% CI: 9–12 Pg; 9%) for temperate, and 8 Pg (95% CI: 6–9 Pg; 11%) for boreal forests.

**Extended Data Table 1 | XGBoost model performance measured using a spatial leave-one-out approach, as described in the Methods section**

| RMSE | R² | MAE | Spatial buffer size (radius in km) | Number of grid cells excluded from the training dataset |
|------|------|-------|------|------|
| 14.41 | 0.61 | 10.73 | 200 | 11 |
| 15.95 | 0.52 | 12.06 | 500 | 75 |
| 17.16 | 0.44 | 13.09 | 1000 | 261 |
| 18.57 | 0.35 | 14.25 | 2000 | 844 |
| 19.68 | 0.28 | 15.14 | 5000 | 2991 |

Root Mean Squared Error (RMSE), R-squared ($R^2$), and Mean Absolute Error (MAE) metrics were calculated using various spatial buffers on the test dataset. The test dataset consisted of a randomly selected 20% subset of the full, original data; however, each test point was evaluated individually using a spatially buffered leave-one-out cross-validation approach. Specifically, for each test point, a model was trained after removing all nearby data points within a specified spatial buffer. The mean number of the nearest grid cells from each test grid cell that were not used to train the model (*that is* excluded by the spatial buffer) is shown in 'Number of grid cells excluded from the training dataset'.

**Extended Data Table 2 | Environmental covariates used to assess drivers of global forest edge effects**

| Covariate name | Source | Original spatial resolution (Arcsecond) | Units | Citation motivating the choice to investigate this covariate | Description |
|---|---|---|---|---|---|
| Mean annual temperature | Karger et al.[40] | 30 | ºC × 10 | Chaplin-Kramer et al.[19]; Reinmann & Hutyra[50]; Smith et al.[6] | Annual climate average |
| Mean annual precipitation | Karger et al.[40] | 30 | mm | Chaplin-Kramer et al.[19]; Smith et al.[6] | |
| Mean annual wind speed | Fick & Hijmans[67] | 30 | m s⁻¹ | Davies-Colley et al.[9]; Smith et al.[6] | |
| Mean soil moisture | Rodell et al.[68] | 360 | kg m⁻² | Chaplin-Kramer et al.[19]; Smith et al.[6] | Soil characteristic |
| Elevation | Amatulli et al.[69] | 30 | m | Chaplin-Kramer et al.[3]; Smith et al.[6] | Topographic factor |
| Slope | Amatulli et al.[69] | 30 | unitless | Smith et al.[6] | |
| Agriculture (Percentage of cultivated and managed vegetation) | Tuanmu & Jetz[70] | 30 | Percentage (0-100) | Van Wilgenburg et al.[71]; Chaplin-Kramer et al.[19]; Smith et al.[6] | Anthropogenic factor |

This table lists the seven environmental variables included in the machine learning analysis, selected based on prior studies[6,9,19,40,50,67–71] suggesting their relevance in shaping edge-related variation in aboveground biomass. For each covariate, the data source, original spatial resolution (in arcseconds), unit of measurement, and key motivating references are provided.
The variables span climatic (for example, temperature, precipitation, wind), edaphic (soil moisture), topographic (elevation, slope), and anthropogenic (agricultural land cover) dimensions.

# Reporting Summary

## Statistics

For all statistical analyses, confirm that the following items are present in the figure legend, table legend, main text, or Methods section.

| n/a | Confirmed | |
|---|---|---|
| ☐ | ☒ | The exact sample size (*n*) for each experimental group/condition, given as a discrete number and unit of measurement |
| ☐ | ☒ | A statement on whether measurements were taken from distinct samples or whether the same sample was measured repeatedly |
| ☐ | ☒ | The statistical test(s) used AND whether they are one- or two-sided *Only common tests should be described solely by name; describe more complex techniques in the Methods section.* |
| ☐ | ☒ | A description of all covariates tested |
| ☐ | ☒ | A description of any assumptions or corrections, such as tests of normality and adjustment for multiple comparisons |
| ☐ | ☒ | A full description of the statistical parameters including central tendency (e.g. means) or other basic estimates (e.g. regression coefficient) AND variation (e.g. standard deviation) or associated estimates of uncertainty (e.g. confidence intervals) |
| ☐ | ☒ | For null hypothesis testing, the test statistic (e.g. *F*, *t*, *r*) with confidence intervals, effect sizes, degrees of freedom and *P* value noted *Give P values as exact values whenever suitable.* |
| ☒ | ☐ | For Bayesian analysis, information on the choice of priors and Markov chain Monte Carlo settings |
| ☒ | ☐ | For hierarchical and complex designs, identification of the appropriate level for tests and full reporting of outcomes |
| ☒ | ☐ | Estimates of effect sizes (e.g. Cohen's *d*, Pearson's *r*), indicating how they were calculated |

*Our web collection on statistics for biologists contains articles on many of the points above.*

## Software and code

Policy information about availability of computer code

| Data collection | Data was downloaded from publicly available sources cited in the manuscript. Citations are provided in the Methods |
|---|---|
| Data analysis | The analyses were run in R version 4.2.1 and Google Earth Engine (https://earthengine.google.com). The code used for this study is available at https://doi.org/10.6084/m9.figshare.29425154 under a Creative Commons licence (CC BY 4.0). |

For manuscripts utilizing custom algorithms or software that are central to the research but not yet described in published literature, software must be made available to editors and reviewers. We strongly encourage code deposition in a community repository (e.g. GitHub). See the Nature Portfolio guidelines for submitting code & software for further information.

## Data

Policy information about availability of data

All manuscripts must include a data availability statement. This statement should provide the following information, where applicable:
- Accession codes, unique identifiers, or web links for publicly available datasets
- A description of any restrictions on data availability
- For clinical datasets or third party data, please ensure that the statement adheres to our policy

The dataset of eight million forested locations, including distance to forest edge and corresponding aboveground biomass density, is publicly available at https://doi.org/10.6084/m9.figshare.29425154 under a Creative Commons licence (CC BY 4.0).

# Research involving human participants, their data, or biological material

Policy information about studies with human participants or human data. See also policy information about sex, gender (identity/presentation), and sexual orientation and race, ethnicity and racism.

| | |
|---|---|
| Reporting on sex and gender | n/a |
| Reporting on race, ethnicity, or other socially relevant groupings | n/a |
| Population characteristics | n/a |
| Recruitment | n/a |
| Ethics oversight | n/a |

Note that full information on the approval of the study protocol must also be provided in the manuscript.

# Field-specific reporting

Please select the one below that is the best fit for your research. If you are not sure, read the appropriate sections before making your selection.

☐ Life sciences      ☐ Behavioural & social sciences      ☒ Ecological, evolutionary & environmental sciences

For a reference copy of the document with all sections, see nature.com/documents/nr-reporting-summary-flat.pdf

# Ecological, evolutionary & environmental sciences study design

All studies must disclose on these points even when the disclosure is negative.

| | |
|---|---|
| Study description | This study measures edge effects on aboveground forest biomass |
| Research sample | 7,837,233 point samples where each point contains distance to the forest edge and forest biomass values |
| Sampling strategy | Random sampling |
| Data collection | We used publicly available datasets, detailed in the Methods. We overlaid a grid on the global forest area and sampled 500 points randomly per grid cell using Google Earth Engine. We exported sampled points in .csv file. |
| Timing and spatial scale | Samples are for the year 2000 and at a global scale |
| Data exclusions | Small samples with less than 20 points per grid cell were excluded. |
| Reproducibility | We have made our code available via a Figshare repository (https://doi.org/10.6084/m9.figshare.29425154). |
| Randomization | n/a |
| Blinding | n/a |

Did the study involve field work?      ☐ Yes      ☒ No

# Reporting for specific materials, systems and methods

We require information from authors about some types of materials, experimental systems and methods used in many studies. Here, indicate whether each material, system or method listed is relevant to your study. If you are not sure if a list item applies to your research, read the appropriate section before selecting a response.

## Materials & experimental systems

| n/a | Involved in the study |
|-----|----------------------|
| ☒ ☐ | Antibodies |
| ☒ ☐ | Eukaryotic cell lines |
| ☒ ☐ | Palaeontology and archaeology |
| ☒ ☐ | Animals and other organisms |
| ☒ ☐ | Clinical data |
| ☒ ☐ | Dual use research of concern |
| ☒ ☐ | Plants |

## Methods

| n/a | Involved in the study |
|-----|----------------------|
| ☒ ☐ | ChIP-seq |
| ☒ ☐ | Flow cytometry |
| ☒ ☐ | MRI-based neuroimaging |

## Plants

| | |
|---|---|
| Seed stocks | n/a |
| Novel plant genotypes | n/a |
| Authentication | n/a |

