## [Peer Review File · Nature Ecology & Evolution]

A globally consistent negative effect of edge on aboveground forest biomass

Corresponding Author: Ms Gayoung Yang

Version 0:

Reviewer comments:

Reviewer #1

(Remarks to the Author)

Review of "A globally consistent negative effect of edge on aboveground forest biomass"

This study analyzed global forest edge effects using a lidar-based aboveground biomass (AGB) product and quantified the impact of forest distance to the edge on AGB. The authors found that 97% of the study areas showed negative impacts from forest fragmentation, and these impacts varied across space, controlled by temperature, agricultural land, and precipitation. The authors also estimated potential reductions in AGB due to edge effects globally. The manuscript is generally well-written, and the figures are well-organized.

However, I have major concerns related to the methodology. My first concern is the potential overestimation of edge effects using a spatial linear regression which only accounts for distance to the edge and spatial autocorrelation while ignoring other factors. The other concern is the neglect of non-static relationships between distance to the edge and biomass loss.

Major Points:

1. Simplified Identification of Edge Effects

Different types of disturbances can exist within 1 km of a forest edge. The authors simplify the identification of edge effects by using distance and accounting only for forest spatial autocorrelation and distance to the edge in the spatial linear model. This approach risks misattributing other disturbances (e.g., those related to agricultural cultivation or selective logging) as "edge effects." Additionally, local climate extremes may also be misclassified as edge effects since the spatial linear regression does not account for disturbance caused by climate extremes that happened in some forests within the edge distance. The authors need to exercise caution in attributing these effects and separating the pure edge effects.

2. Static Assumptions in Linear Modeling

The spatial linear regression model quantifies edge effects by providing magnitudes and signs of their impact on biomass. However, the effect of distance to the edge diminishes with increasing distance, meaning that the impact near the edge (e.g., within a few meters) is disproportionately higher than at longer distances. A static linear model cannot accurately represent these dynamics. The authors should consider using a non-linear model or another approach to better capture this proportional change which cannot be represented even the authors tested spearman correlation.

3. Unclear Mechanisms for Precipitation-Related Effects

The relationship between precipitation and edge effects, as shown in Figure 2d, is unclear. The authors hypothesize that drought-adaptive plants and water conservation mechanisms play a role, but the underlying mechanisms remain ambiguous. For example, increased turbulence at forest edges might simultaneously increase precipitation, temperature, and vapor pressure deficit (VPD) can increase due to decreased ET, making it challenging to isolate the effects of water stress. If VPD is the main driver, the authors should include it in the analysis rather than relying on speculative assumptions.

Minor points:

1. The main findings related to climate and agricultural impacts on the forest edge effects should also be tested with tree cover data, to reduce the potential bias using one AGB product.
2. Line 129: The data source and details about the agriculture data are not introduced in any part of the manuscript.
3. Line 138: Leave-one-out is confusing, as in the method, the authors introduced about 20% random samples are removed for the calculation of R².
4. Line 365: Why biome-level XGBoost models are needed where a global-generic model can already work on understanding factors of edge effects?
5. Figure 2: Why there are different SHAP values in e-f compared to d? Same SHAP values should be presented regarding contributions from mean annual precipitation.

Reviewer #2

(Remarks to the Author)

Thank you for this interesting study. I really enjoyed reading your article, which is well structured and uses advanced statistical modelling to answer the proposed research questions. Apart from a few major comments (described below), I have also a number of text edits, additional literature and raised a few questions with respect to clarity of some parts. Overall, however, this is a very relevant and potentially impactful study which further underlines the detrimental interaction between habitat fragmentation and climate change. The authors call for a change in policy and management and for the consideration of edge effects, which is timely and aligns well with previous recent literature.

Larger comments:

The introduction is complete and addresses the state of the art and knowledge gaps profoundly. However, I was missing some potentially relevant literature at some points, which I added in my comments.

The results section is quite extensive and some parts can be removed to increase the focus of the story line. I also noticed that the results are already intermixed with discussions. I actually like this way of writing, but then I found it somewhat strange that you have a separate discussion section. I suggest to choose one of both approaches, but potentially it is also limited by the journal's guidelines.

The methodology is quite complex and I have to admit that I was not familiar with all modelling methods. However, I found that the authors described the rather advanced methodology in a relatively logical way, which was much appreciated. The only larger concern I had with the applied methodology was the use of relative "old" maps from Hansen et al. and Harris et al. I was wondering if there is not a more recent source available at a comparable resolution, and what effect this could have on your outcomes. I think the authors could better justify the use of these older maps and discuss this in the main text. Do you expect that the situation now is the same as in 2000, or could there be differences in your findings? Recent field-based measurements in 9-m radius plots from across Europe, for instance, find an opposite pattern (higher biomass in the edges). Of course, the scale of these studies is entirely different, but still I think this requires some deeper discussion.

Finally, fine-scale drivers such as soil parameters or forest microclimate were not included here, which is fine because I do not think this data is already available at a global extent. Yet, these drivers, and especially microclimate, can vary quite markedly from edge to core and have a strong influence in forests. Maybe this could also be discussed a bit more.

Minor suggestions:

L7: I suggest to use: Direction and magnitude of edge effects

L9: Please add for clarity: "on biomass" after edge effect

L12: I suggest to use past tense: were instead of are

L24: I think it is just km instead of km²?

L24: please move ' before comma

L24-25: There are also edge-to-interior gradients in edaphic conditions. We have actually very recently published some articles around these topics: <https://doi.org/10.1111/jvs.12844> <https://doi.org/10.1016/j.agrformet.2021.108699> <https://doi.org/10.1038/s41559-024-02335-6> These could potentially be relevant for the intro?

L39: The additional literature that I have provided above, could actually also be very relevant here. It's not global, but already a generalization across the European temperate biome.

L61: direction instead of sign?

L69: Hansen et al., I was just wondering if there is not a more recent source available as this map already dates from 2000?

The EU provides global forest cover maps for 2020, but perhaps at a much coarser resolution

(https://forobs.jrc.ec.europa.eu/GF_C#data_download). You also have the ESA world cover map (<https://esa-worldcover.org/en>), which includes forest cover. Maybe recent satellite or Lidar maps (I'm not an expert on this topic) could provide a better resolution and more recent view of the situation?

L80: please add i.e. before negative (part between brackets)

L106: Variable or driver sounds more logical to me than "feature"

L107: arranged instead of aligned?

L115-118: I'm not sure that I understand this subfigure g completely. Is agriculture also included? Could you clarify?

L128-130: I found this sentence rather complex with too many commas. Could you rephrase?

L148-154: There is already a lot of discussion in this part, intermixed with results. Of course this is fine but then I was confused that you have an additional discussion section below.

L156-163: also a lot of discussion

L165-174: discussion

L185-193: I have the feeling these biome level distinctions were already discussed in the parts above. I wonder if it is really necessary to repeat it separately here.

L197: The figure legend in (a) seems to show Tg, but in (b) Pg is mentioned. Could you clarify?

L213: 58 Pg, could you add uncertainty (eg SD)?

L229-231: I think this needs a bit deeper discussion. There is indeed a strong discrepancy between your findings and the ones by Meeussen et al. in Science of the Total Environment. Of course, I agree there is also a huge difference in scale, with Meeussen et al. using ground-based TLS to quantify plot-scale (9 m radius) AGB, while you are using globally gridded data. Still, Meeussen et al. also uses a continuous edge-interior gradient of 100m from different locations with a wide range of macroclimates and landscape contexts. They reported a consistent pattern of AGB being 95% higher in edges than interiors. So, I was wondering how the field-based data replicated at continental scale and global data still give opposite patterns. Could you elaborate?

L249-250: Please change to a less strong wording, e.g. is in line with or corroborates

L272-276: Very convincing conclusion!

L286-287: Is this the most recent source available? If yes, I would highlight that in the text and write that it is the best available product at this moment and at this fine resolution. What would you expect from using a more recent map? How could this impact your results? Maybe shortly elaborate on that, potentially even in the main text.

L304: describe

L321: mile, please consider SI units

L343: I have to be honest here, but I'm not at all an expert on this type of model. However, I feel that the authors have quite nicely explained the complex workflow, which was highly appreciated.

L345-346: Could you briefly repeat the different tested drivers here for clarity? I was also wondering what the effect of small-scale driver would be, such as some soil parameters or even forest microclimate. I feel this requires a few lines of discussion in the main text. In our recent studies (<https://doi.org/10.1038/s41559-024-02335-6>) we found that these fine-scale drivers (esp. microclimate) can actually have quite strong effects.

L363: Please add equal to or = after RMSE (also for following values).

L366: Please add: separately after forests.

L372: Idem here. Please add equal signs before values for clarity.

L378: Feature, maybe this term is more related to the type of model you are using, but in my opinion "variable" sounds a bit more fitting. It's also what I'm used to from literature. I think variables is also the more commonly used term in traditional statistics.

L396: "To do this". I suggest to change this to: To this end

L402: "Edge limit". In traditional forest edge literature, I think this is called "depth of the edge effect".

L410: please change to "if the edge limit was equal to 200 m"

L411: I suggest to change to "within 200 m from the nearest edge"

L415: Maybe you could also convert this amount to carbon stocks in order to quantify the actual impact on the global C balance?

I have also added my text edits and minor comments in the attached PDF file.

Good luck with the review! I already look forward to the edited version.

Kind regards,
Thomas Vanneste

Reviewer #3

(Remarks to the Author)

This is an important paper at two scales, the scale of biomes and the scale of the globe. The global detail and rigor of this analysis brings into sharp focus the knock-on effects of habitat loss and fragmentation on ecosystems and carbon lost from ecological systems. Because such a high proportion of forest is near the edge, this study finds that the impacts of habitat fragmentation on carbon lost is massive. At the scale of biomes, this study resolves a pattern that has, in the past, proven inconsistent: variation in edge effects on aboveground biomass near, for example, edges of tropical versus temperate forests. Discrepancies have hindered understanding and conservation attention in this area. This new analysis, with unprecedented extent and (at that extent) precision points to the generality of the loss of biomass near habitat edges. With regard to edge effects, I'd like to see more depth in explanation of the following three questions.

1. What explains the "short" distance of edge effects? Working with data from tropical forests, Chaplin-Kramer, et al. (2019) that you cited shows edge effects that extend much deeper into the forest. (this paper deserves to be cited Broadbent, et al. 2008 see Fig. 6)

2. What explains the difference between these results of consistent negative edge effects across biomes, and other studies have found a different response? I found the explanations given to be interesting, but superficial. In fact, these and past results are consistent in one respect: the edge effect on biomass is much lower in the temperate than in tropical forest; perhaps this change in magnitude has been confused for change in direction of response?

3. What is the mechanism of temperature effect on reduced aboveground biomass? I ask because there are two possible temperature effects that may be interacting. The authors focus on climate change. Yet, it is also well known that edges themselves affect (and can increase) temperatures (ie, Tuff, et al. 2016). What is the impact of environmental versus edge-caused temperature change?

I do not work with remotely sensed data, and especially not at regional and global scales. Thus, I am not able to evaluate analyses in this paper. However, I have collaborated with such scientists and the data are appropriate for such global analyses.

Tuff, K.T., Tuff, T. and Davies, K.F., 2016. A framework for integrating thermal biology into fragmentation research. *Ecology letters*, 19(4), pp.361-374.

Broadbent, E.N., Asner, G.P., Keller, M., Knapp, D.E., Oliveira, P.J. and Silva, J.N., 2008. Forest fragmentation and edge effects from deforestation and selective logging in the Brazilian Amazon. *Biological conservation*, 141(7), pp.1745-1757.

Decision Letter:

11th February 2025

Dear Dr Yang,

Your Article, "A globally consistent negative effect of edge on aboveground forest biomass" has now been seen by 3 reviewers. You will see from their comments copied below that while they find your work of considerable potential interest, they have raised quite substantial concerns that must be addressed. In light of these comments, we cannot accept the manuscript for publication, but would be happy to consider a revised version that addresses these serious concerns.

We hope you will find the reviewers' comments useful as you decide how to proceed. If you wish to submit a substantially revised manuscript, please bear in mind that we will be reluctant to approach the reviewers again in the absence of major revisions.

If you choose to revise your manuscript taking into account all reviewer and editor comments, please highlight all changes in the manuscript text file.

* Include a "Response to reviewers" document detailing, point-by-point, how you addressed each referee comment. If no action was taken to address a point, you must provide a compelling argument. This response will be sent back to the referees along with the revised manuscript.

* If you have not done so already we suggest that you begin to revise your manuscript so that it conforms to our Article format instructions at <http://www.nature.com/natecolevol/info/final-submission>. Refer also to any guidelines provided in this letter.

Link Redacted

If you wish to submit a suitably revised manuscript we would hope to receive it within 6 months. If you cannot send it within this time, please let us know. We will be happy to consider your revision so long as nothing similar has been accepted for publication at Nature Ecology & Evolution or published elsewhere.

Nature Ecology & Evolution is committed to improving transparency in authorship. As part of our efforts in this direction, we are now requesting that all authors identified as 'corresponding author' on published papers create and link their Open Researcher and Contributor Identifier (ORCID) with their account on the Manuscript Tracking System (MTS), prior to acceptance. This applies to primary research papers only. ORCID helps the scientific community achieve unambiguous attribution of all scholarly contributions. You can create and link your ORCID from the home page of the MTS by clicking on 'Modify my Springer Nature account'. For more information please visit www.springernature.com/orcid.

Thank you for the opportunity to review your work.

[Redacted]

Reviewer expertise:

Reviewer #1: vegetation dynamics, machine learning, remote-sensed data

Reviewer #2: vegetation responses to disturbance, edge effects

Reviewer #3: habitat fragmentation

Reviewers' comments:

Reviewer #1 (Remarks to the Author):

Review of "A globally consistent negative effect of edge on aboveground forest biomass"

This study analyzed global forest edge effects using a lidar-based aboveground biomass (AGB) product and quantified the impact of forest distance to the edge on AGB. The authors found that 97% of the study areas showed negative impacts from forest fragmentation, and these impacts varied across space, controlled by temperature, agricultural land, and precipitation. The authors also estimated potential reductions in AGB due to edge effects globally. The manuscript is generally well-written, and the figures are well-organized.

However, I have major concerns related to the methodology. My first concern is the potential overestimation of edge effects using a spatial linear regression which only accounts for distance to the edge and spatial autocorrelation while ignoring other factors. The other concern is the neglect of non-static relationships between distance to the edge and biomass loss.

Major Points:

1. Simplified Identification of Edge Effects

Different types of disturbances can exist within 1 km of a forest edge. The authors simplify the identification of edge effects by using distance and accounting only for forest spatial autocorrelation and distance to the edge in the spatial linear model. This approach risks misattributing other disturbances (e.g., those related to agricultural cultivation or selective logging) as "edge effects." Additionally, local climate extremes may also be misclassified as edge effects since the spatial linear regression does not account for disturbance caused by climate extremes that happened in some forests within the edge distance. The authors need to exercise caution in attributing these effects and separating the pure edge effects.

2. Static Assumptions in Linear Modeling

The spatial linear regression model quantifies edge effects by providing magnitudes and signs of their impact on biomass. However, the effect of distance to the edge diminishes with increasing distance, meaning that the impact near the edge (e.g., within a few meters) is disproportionately higher than at longer distances. A static linear model cannot accurately represent these dynamics. The authors should consider using a non-linear model or another approach to better capture this proportional change which cannot be represented even the authors tested spearman correlation.

3. Unclear Mechanisms for Precipitation-Related Effects

The relationship between precipitation and edge effects, as shown in Figure 2d, is unclear. The authors hypothesize that drought-adaptive plants and water conservation mechanisms play a role, but the underlying mechanisms remain ambiguous. For example, increased turbulence at forest edges might simultaneously increase precipitation, temperature, and vapor pressure deficit (VPD) can increase due to decreased ET, making it challenging to isolate the effects of water stress. If VPD is the main driver, the authors should include it in the analysis rather than relying on speculative assumptions.

Minor points:

1. The main findings related to climate and agricultural impacts on the forest edge effects should also be tested with tree cover data, to reduce the potential bias using one AGB product.
2. Line 129: The data source and details about the agriculture data are not introduced in any part of the manuscript.
3. Line 138: Leave-one-out is confusing, as in the method, the authors introduced about 20% random samples are removed for the calculation of R².
4. Line 365: Why biome-level XGBoost models are needed where a global-generic model can already work on understanding factors of edge effects?
5. Figure 2: Why there are different SHAP values in e-f compared to d? Same SHAP values should be presented regarding contributions from mean annual precipitation.

Reviewer #2 (Remarks to the Author):

Thank you for this interesting study. I really enjoyed reading your article, which is well structured and uses advanced statistical modelling to answer the proposed research questions. Apart from a few major comments (described below), I have also a number of text edits, additional literature and raised a few questions with respect to clarity of some parts. Overall, however, this is a very relevant and potentially impactful study which further underlines the detrimental interaction between habitat fragmentation and climate change. The authors call for a change in policy and management and for the consideration of edge effects, which is timely and aligns well with previous recent literature.

Larger comments:

The introduction is complete and addresses the state of the art and knowledge gaps profoundly. However, I was missing some potentially relevant literature at some points, which I added in my comments.

The results section is quite extensive and some parts can be removed to increase the focus of the story line. I also noticed that the results are already intermixed with discussions. I actually like this way of writing, but then I found it somewhat strange that you have a separate discussion section. I suggest to choose one of both approaches, but potentially it is also limited by the journal's guidelines.

The methodology is quite complex and I have to admit that I was not familiar with all modelling methods. However, I found that the authors described the rather advanced methodology in a relatively logical way, which was much appreciated. The only larger concern I had with the applied methodology was the use of relative "old" maps from Hansen et al. and Harris et al. I was wondering if there is not a more recent source available at a comparable resolution, and what effect this could have

on your outcomes. I think the authors could better justify the use of these older maps and discuss this in the main text. Do you expect that the situation now is the same as in 2000, or could there be differences in your findings? Recent field-based measurements in 9-m radius plots from across Europe, for instance, find an opposite pattern (higher biomass in the edges). Of course, the scale of these studies is entirely different, but still I think this requires some deeper discussion.

Finally, fine-scale drivers such as soil parameters or forest microclimate were not included here, which is fine because I do not think this data is already available at a global extent. Yet, these drivers, and especially microclimate, can vary quite markedly from edge to core and have a strong influence in forests. Maybe this could also be discussed a bit more.

Minor suggestions:

L7: I suggest to use: Direction and magnitude of edge effects

L9: Please add for clarity: "on biomass" after edge effect

L12: I suggest to use past tense: were instead of are

L24: I think it is just km instead of km²?

L24: please move ' before comma

L24-25: There are also edge-to-interior gradients in edaphic conditions We have actually very recently published some articles around these topics: <https://doi.org/10.1111/jvs.12844> <https://doi.org/10.1016/j.agrformet.2021.108699> <https://doi.org/10.1038/s41559-024-02335-6> These could potentially be relevant for the intro?

L39: The additional literature that I have provided above, could actually also be very relevant here. It's not global, but already a generalization across the European temperate biome.

L61: direction instead of sign?

L69: Hansen et al., I was just wondering if there is not a more recent source available as this map already dates from 2000?

The EU provides global forest cover maps for 2020, but perhaps at a much coarser resolution

(https://forobs.jrc.ec.europa.eu/GF_C#data_download). You also have the ESA world cover map (<https://esa-worldcover.org/en>), which includes forest cover. Maybe recent satellite or Lidar maps (I'm not an expert on this topic) could provide a better resolution and more recent view of the situation?

L80: please add i.e. before negative (part between brackets)

L106: Variable or driver sounds more logical to me than "feature"

L107: arranged instead of aligned?

L115-118: I'm not sure that I understand this subfigure g completely. Is agriculture also included? Could you clarify?

L128-130: I found this sentence rather complex with too many commas. Could you rephrase?

L148-154: There is already a lot of discussion in this part, intermixed with results. Of course this is fine but then I was confused that you have an additional discussion section below.

L156-163: also a lot of discussion

L165-174: discussion

L185-193: I have the feeling these biome level distinctions were already discussed in the parts above. I wonder if it is really necessary to repeat it separately here.

L197: The figure legend in (a) seems to show Tg, but in (b) Pg is mentioned. Could you clarify?

L213: 58 Pg, could you add uncertainty (eg SD)?

L229-231: I think this needs a bit deeper discussion. There is indeed a strong discrepancy between your findings and the ones by Meeussen et al. in Science of the Total Environment. Of course, I agree there is also a huge difference in scale, with Meeussen et al. using ground-based TLS to quantify plot-scale (9 m radius) AGB, while you are using globally gridded data. Still, Meeussen et al. also uses a continuous edge-interior gradient of 100m from different locations with a wide range of macroclimates and landscape contexts. They reported a consistent pattern of AGB being 95% higher in edges than interiors. So, I was wondering how the field-based data replicated at continental scale and global data still give opposite patterns. Could you elaborate?

L249-250: Please change to a less strong wording, e.g. is in line with or corroborates

L272-276: Very convincing conclusion!

L286-287: Is this the most recent source available? If yes, I would highlight that in the text and write that it is the best available product at this moment and at this fine resolution. What would you expect from using a more recent map? How could this impact your results? Maybe shortly elaborate on that, potentially even in the main text.

L304: describe

L321: mile, please consider SI units

L343: I have to be honest here, but I'm not at all an expert on this type of model. However, I feel that the authors have quite nicely explained the complex workflow, which was highly appreciated.

L345-346: Could you briefly repeat the different tested drivers here for clarity? I was also wondering what the effect of small-scale driver would be, such as some soil parameters or even forest microclimate. I feel this requires a few lines of discussion in the main text. In our recent studies (<https://doi.org/10.1038/s41559-024-02335-6>) we found that these fine-scale drivers (esp. microclimate) can actually have quite strong effects.

L363: Please add equal to or = after RMSE (also for following values).

L366: Please add: separately after forests.

L372: Idem here. Please add equal signs before values for clarity.

L378: Feature, maybe this term is more related to the type of model you are using, but in my opinion "variable" sounds a bit more fitting. It's also what I'm used to from literature. I think variables is also the more commonly used term in traditional statistics.

L396: "To do this". I suggest to change this to: To this end

L402: "Edge limit". In traditional forest edge literature, I think this is called "depth of the edge effect".

L410: please change to "if the edge limit was equal to 200 m"

L411: I suggest to change to "within 200 m from the nearest edge"

L415: Maybe you could also convert this amount to carbon stocks in order to quantify the actual impact on the global C balance?

I have also added my text edits and minor comments in the attached PDF file.

Good luck with the review! I already look forward to the edited version.

Kind regards,
Thomas Vanneste

Reviewer #3 (Remarks to the Author):

This is an important paper at two scales, the scale of biomes and the scale of the globe. The global detail and rigor of this analysis brings into sharp focus the knock-on effects of habitat loss and fragmentation on ecosystems and carbon lost from ecological systems. Because such a high proportion of forest is near the edge, this study finds that the impacts of habitat fragmentation on carbon lost is massive. At the scale of biomes, this study resolves a pattern that has, in the past, proven inconsistent: variation in edge effects on aboveground biomass near, for example, edges of tropical versus temperate forests. Discrepancies have hindered understanding and conservation attention in this area. This new analysis, with unprecedented extent and (at that extent) precision points to the generality of the loss of biomass near habitat edges. With regard to edge effects, I'd like to see more depth in explanation of the following three questions.

1. What explains the "short" distance of edge effects? Working with data from tropical forests, Chaplin-Kramer, et al. (2019) that you cited shows edge effects that extend much deeper into the forest. (this paper deserves to be cited Broadbent, et al. 2008 see Fig. 6)

2. What explains the difference between these results of consistent negative edge effects across biomes, and other studies have found a different response? I found the explanations given to be interesting, but superficial. In fact, these and past results are consistent in one respect: the edge effect on biomass is much lower in the temperate than in tropical forest; perhaps this change in magnitude has been confused for change in direction of response?

3. What is the mechanism of temperature effect on reduced aboveground biomass? I ask because there are two possible temperature effects that may be interacting. The authors focus on climate change. Yet, it is also well known that edges themselves affect (and can increase) temperatures (ie, Tuff, et al. 2016). What is the impact of environmental versus edge-caused temperature change?

I do not work with remotely sensed data, and especially not at regional and global scales. Thus, I am not able to evaluate analyses in this paper. However, I have collaborated with such scientists and the data are appropriate for such global analyses.

Tuff, K.T., Tuff, T. and Davies, K.F., 2016. A framework for integrating thermal biology into fragmentation research. *Ecology letters*, 19(4), pp.361-374.

Broadbent, E.N., Asner, G.P., Keller, M., Knapp, D.E., Oliveira, P.J. and Silva, J.N., 2008. Forest fragmentation and edge effects from deforestation and selective logging in the Brazilian Amazon. *Biological conservation*, 141(7), pp.1745-1757.

Version 1:

Reviewer comments:

Reviewer #1

(Remarks to the Author)

The authors addressed the concerns I wrote about in my previous comments. I agree that a more concrete understanding of the mechanism can be encouraged for future work, and the authors included this in the discussion. It's also encouraging to see the code and generated data on forest edge distance, loss of biomass, and sensitivity to distance in the final publication. I recommend that the authors also share the full data processing scripts from start to finish, which would benefit the community.

Reviewer #2

(Remarks to the Author)

In my opinion, the authors have done an exceptionally good job providing highly adequate and convincing responses to the concerns raised in my previous review. Therefore, my remaining comments are only minor (I supply them as in-text edits in the PDF file attached). Hopefully, you find these remaining comments helpful and after implementation, I think this study is completely ready for publication.

After reading this revised version and the convincing reasoning in the rebuttal letter, I now understand why the 2000 maps were used and not a more recent (real-time) estimation of the forest cover and associated biomass situation. Yet, I still wonder a bit what the impact of a more recent map would be on your results. Will the situation have aggravated and differences between edge and interior biomass become even more pronounced? Or, at least for the situation here in Flanders and probably elsewhere in central Europe, forests have become so heavily fragmented that interiors virtually not exist any longer. Almost everything would be quantified as edge from a fine-scale (e.g., microclimate) point of view. In other words, differences between edges and edge-like interiors would not be much different.

Anyway, perhaps this goes to far and is more applicable to finer scales whereas your analyses are global. Good luck with the final revisions and already a big congratulations on the impressive work.

Best regards,
Thomas Vanneste

Reviewer #3

(Remarks to the Author)

The authors have successfully addressed my and other reviewers' comments.

Decision Letter:

24th June 2025

Dear Dr. Yang,

Thank you for submitting your revised manuscript "A globally consistent negative effect of edge on aboveground forest biomass" (NATECOLEVOL-24123528A). It has now been seen again by the original reviewers and their comments are below. The reviewers find that the paper has improved in revision, and therefore we'll be happy in principle to publish it in Nature Ecology & Evolution, pending minor revisions to satisfy the reviewers' final suggestions (including on the attached document) and to comply with our editorial and formatting guidelines.

If you have not done so already, please ensure that you also email us a completed copy of the Reporting summary :

Reporting summary: https://www.nature.com/documents/nr-reporting-summary.pdf

[Redacted]

Reviewer #1 (Remarks to the Author):

The authors addressed the concerns I wrote about in my previous comments. I agree that a more concrete understanding of the mechanism can be encouraged for future work, and the authors included this in the discussion. It's also encouraging to see the code and generated data on forest edge distance, loss of biomass, and sensitivity to distance in the final publication. I recommend that the authors also share the full data processing scripts from start to finish, which would benefit the community.

Reviewer #2 (Remarks to the Author):

In my opinion, the authors have done an exceptionally good job providing highly adequate and convincing responses to the concerns raised in my previous review. Therefore, my remaining comments are only minor (I supply them as in-text edits in the PDF file attached). Hopefully, you find these remaining comments helpful and after implementation, I think this study is completely ready for publication.

After reading this revised version and the convincing reasoning in the rebuttal letter, I now understand why the 2000 maps were used and not a more recent (real-time) estimation of the forest cover and associated biomass situation. Yet, I still wonder a bit what the impact of a more recent map would be on your results. Will the situation have aggravated and differences between edge and interior biomass become even more pronounced? Or, at least for the situation here in Flanders and probably elsewhere in central Europe, forests have become so heavily fragmented that interiors virtually not exist any longer. Almost everything would be quantified as edge from a fine-scale (e.g., microclimate) point of view. In other words, differences between edges and edge-like interiors would not be much different.

Anyway, perhaps this goes to far and is more applicable to finer scales whereas your analyses are global. Good luck with the final revisions and already a big congratulations on the impressive work.

Best regards,
Thomas Vanneste

Reviewer #3 (Remarks to the Author):

The authors have successfully addressed my and other reviewers' comments.

Submission ID NATECOLEVOL-24123528

Title: A globally consistent negative effect of edge on aboveground forest biomass

Journal: Nature Ecology and Evolution

Reviewer #1

“Review of “A globally consistent negative effect of edge on aboveground forest biomass”

This study analyzed global forest edge effects using a lidar-based aboveground biomass (AGB) product and quantified the impact of forest distance to the edge on AGB. The authors found that 97% of the study areas showed negative impacts from forest fragmentation, and these impacts varied across space, controlled by temperature, agricultural land, and precipitation. The authors also estimated potential reductions in AGB due to edge effects globally. The manuscript is generally well-written, and the figures are well-organized.

However, I have major concerns related to the methodology. My first concern is the potential overestimation of edge effects using a spatial linear regression which only accounts for distance to the edge and spatial autocorrelation while ignoring other factors. The other concern is the neglect of non-static relationships between distance to the edge and biomass loss.”

Response 1: We sincerely thank the reviewer for their thoughtful and constructive feedback on our manuscript. We appreciate the time and effort you invested in carefully evaluating our work and for highlighting two important methodological concerns. Below, we respond to each of your concerns in detail and describe how we have addressed them in the revised manuscript.

The first concern refers to a potential overestimation of edge effects due to the spatial regression model accounting only for distance to edge and spatial autocorrelation, while excluding other factors. We understand and appreciate the concern that other factors, such as agricultural cultivation and climate extremes, could be misattributed to 'edge effects' in our analysis. However, our study is designed to quantify the net realized effect of proximity to forest edge on aboveground biomass, irrespective of the underlying mechanisms. In this context, the term 'edge effect' refers to the observed gradient in biomass from edge to interior, capturing the aggregate outcome of all direct and indirect influences associated with edge environments.

We agree that if our objective had been to isolate the causal influence of edge creation per se, controlling for other covariates would be necessary. However, our aim was to empirically describe the general global pattern of biomass variation with distance from edge, without constraining the sources of that variation. This approach is widely used in ecological edge effect studies and aligns with prior work (e.g., Chaplin-Kramer et al. 2015). To address drivers of variation in the strength of edge effects across space, we complemented our regression analysis with machine learning (XGBoost) and SHAP analysis, which incorporated climate, land use, and topographic variables to explain spatial differences in edge effect magnitude. We now explain in L56-58 that in our study edge effects refer to the overall impact of edge on biomass.

The second concern refers to the potential neglect of the non-static (i.e., diminishing) relationship between edge proximity and biomass loss. We fully agree that the influence of edge effects is not constant across space and diminishes with increasing distance from the edge. To reflect this, we employed a spatial log-linear regression model, where distance to edge is log₁₀-transformed. This explicitly captures the expected non-linearity of edge effects, with a stronger impact near the edge and a gradual effect decline with distance. To clarify this in the manuscript, we have revised all instances of 'spatial linear regression' to 'spatial log-linear regression', and added a brief explanation of why this model structure was chosen to reflect the expected shape of edge influence (see L361-363).

“Major Points:

1. Simplified Identification of Edge Effects

Different types of disturbances can exist within 1 km of a forest edge. The authors simplify the identification of edge effects by using distance and accounting only for forest spatial autocorrelation and distance to the edge in the spatial linear model. This approach risks misattributing other disturbances (e.g., those related to agricultural cultivation or selective logging) as “edge effects.” Additionally, local climate extremes may also be misclassified as edge effects since the spatial linear regression does not account for disturbance caused by climate extremes that happened in some forests within the edge distance. The authors need to exercise caution in attributing these effects and separating the pure edge effects.”

Response 2: Thank you for your valuable insights. We fully acknowledge your concern regarding the potential conflation of multiple disturbances—such as agricultural activity, selective logging, or local climate extremes—with what we term “edge effects.” Our study intentionally defines edge effects as the net pattern of AGB variation along the edge-to-interior gradient. This pattern reflects the combined influence of all processes that are spatially structured with respect to forest edges. These include—but are not limited to—altered soil and microclimate, increased exposure to disturbances (e.g., fire, wind, logging), and anthropogenic pressures such as adjacent agricultural land use. In this framework, distance to edge serves as a proxy variable for the cumulative influence of these interacting mechanisms.

We agree that this approach does not attempt to isolate the pure or causal effect of edge creation itself. Instead, it captures the realized ecological outcome of proximity to forest edges. This choice is intentional, as our primary goal is to assess how biomass patterns vary with edge proximity globally, not to disentangle individual drivers. Nevertheless, to explore underlying mechanisms, we complemented our regression analysis with a machine learning model (XGBoost) and SHAP interpretation to identify environmental and anthropogenic drivers that explain variation in edge effects across space.

To clarify this conceptual framing, we have revised the manuscript (L56–58) to explicitly state that the “edge effects” we report reflect the aggregate result of multiple biotic and abiotic influences that tend to co-occur near forest edges. We have also highlighted this point in the main text (L213-216) to avoid any misinterpretation that our study isolates edge effects in a mechanistic or causal sense.

We hope this clarification resolves the concern and improves the transparency of our methodological framework.

“2. Static Assumptions in Linear Modeling

The spatial linear regression model quantifies edge effects by providing magnitudes and signs of their impact on biomass. However, the effect of distance to the edge diminishes with increasing distance, meaning that the impact near the edge (e.g., within a few meters) is disproportionately higher than at longer distances. A static linear model cannot accurately represent these dynamics. The authors should consider using a non-linear model or another approach to better capture this proportional change which cannot be represented even the authors tested spearman correlation.”

Response 3: We fully agree that the impact of edge proximity on biomass is not uniform across space and that the strongest effects typically occur very close to the edge, tapering off with distance into the forest interior. To account for this non-linear behavior, we did not use a strictly linear model. Instead, we employed a spatial log-linear regression, in which distance to edge is log₁₀-transformed. This transformation allows the model to reflect the expected steep decline in biomass near the edge and the progressively weaker influence at greater distances. Thus, our model captures the

proportional and diminishing nature of edge effects, aligning with both ecological theory and empirical findings from prior studies (e.g., Chaplin-Kramer et al. 2015).

In addition, to ensure that the results were not dependent on parametric model assumptions, we conducted a complementary analysis using non-parametric Spearman correlations, which produced qualitatively similar results (Extended Data Fig. 2a). This robustness check supports the validity of our main findings.

We have now revised all instances of “spatial linear regression” to “spatial log-linear regression” to avoid ambiguity and added a brief explanation in the Methods section to justify the choice of log-transformation for modeling the edge-interior biomass gradient (L361-363). We hope this clarification addresses your concern and illustrates how our model accounts for the dynamic nature of edge influence.

“3. Unclear Mechanisms for Precipitation-Related Effects

The relationship between precipitation and edge effects, as shown in Figure 2d, is unclear. The authors hypothesize that drought-adaptive plants and water conservation mechanisms play a role, but the underlying mechanisms remain ambiguous. For example, increased turbulence at forest edges might simultaneously increase precipitation, temperature, and vapor pressure deficit (VPD) can increase due to decreased ET, making it challenging to isolate the effects of water stress. If VPD is the main driver, the authors should include it in the analysis rather than relying on speculative assumptions.”

Response 4: We greatly appreciate your comment regarding the mechanisms underlying precipitation-related edge effects and the concern about the potential ambiguity of our interpretation. In our manuscript, we introduced water stress as a potential explanatory factor because the patterns shown in Figure 2d–2e indicate that edge-related biomass loss tends to increase from dry to more humid regions—suggesting that susceptibility to edge effects may peak in moderately wet forests that are not drought-adapted. This gradient led us to consider water limitation and drought sensitivity as plausible contributing factors, especially given that edge environments experience altered microclimatic conditions that could amplify evapotranspiration stress.

We agree that the relationship between precipitation and edge effects is complex, and we acknowledge that additional factors—such as turbulence-induced changes in temperature and vapor pressure deficit (VPD)—may interact with water availability in ways that complicate interpretation. While we hypothesized that drought-adaptive traits and water conservation mechanisms could influence biomass responses near edges, we recognize that these remain speculative and require more targeted, mechanistic investigation.

As noted in the manuscript, our analysis is primarily descriptive and aims to identify global patterns of edge effects rather than establish direct causal links. Our narrative explanations, which explore drought adaptation, water limitation, and microclimatic changes, are meant to provide ecological context rather than definitive mechanistic conclusions. We fully recognize that these interpretations are preliminary and should not be viewed as conclusive statements on the drivers of precipitation-related edge effects.

In response to your suggestion, we have revised the manuscript to clarify that the observed relationships should be interpreted with caution. Specifically, we have added a statement (L213–216) emphasizing the need for future studies to explicitly test the mechanistic roles of variables such as forest microclimate. These studies would help disentangle the complex interactions between these factors and their contribution to edge effect dynamics. We also note that while VPD could indeed be

an important driver, its inclusion in the analysis requires more detailed and specific data, which is beyond the scope of this study.

We hope this revision addresses your concern and clarifies the intent of our narrative on precipitation-related edge effects.

“Minor points:

1. The main findings related to climate and agricultural impacts on the forest edge effects should also be tested with tree cover data, to reduce the potential bias using one AGB product.”

Response 5: We appreciate the suggestion to incorporate tree cover data as a means of validating our findings and mitigating potential biases inherent in relying on a single above-ground biomass (AGB) product. To address this, we incorporated additional results using an XGBoost model that leverages tree cover data, now presented in Extended Data Figure 9. Overall, the results were consistent with those based on AGB, though some differences were observed in tropical forests and regions with significant agricultural land. We hypothesize that this variation arises because tree canopy cover is not the sole factor influencing biomass. Other factors, such as tree height, crown width, and spatial distribution, also contribute to biomass variation and can differ across regions. For example, recent studies have shown that large trees near tropical forest edges tend to be smaller and thinner than those deeper in the forest, although their crown widths remain similar (Nunes et al., 2023). We have now included a discussion of these findings in a new paragraph (L191-203 and L441-443) to further clarify these points.

Nunes, M. H. et al. Edge effects on tree architecture exacerbate biomass loss of fragmented Amazonian forests. *Nat Commun* 14, 8129 (2023).

“2. Line 129: The data source and details about the agriculture data are not introduced in any part of the manuscript.”

Response 6: Thank you for pointing out the lack of clarity regarding the agricultural data source. The term "agriculture" refers to the "percentage of cultivated and managed vegetation," as introduced in Line 149. To ensure better clarity, we have explicitly reiterated this definition in Line 416. Additionally, we have updated Extended Data Table 1 to provide a clearer description, specifying that the covariate is named "Agriculture (Percentage of cultivated and managed vegetation)."

“3. Line 138: Leave-one-out is confusing, as in the method, the authors introduced about 20% random samples are removed for the calculation of R².”

Response 7: We appreciate your feedback regarding the "leave-one-out" approach in our methodology. To clarify, the test dataset consists of a randomly selected 20% subset of the original data, and for each test point, we trained the model using a dataset from which all nearby data points within a defined spatial buffer were excluded. The R² was then calculated based on the predictions for these test samples. We have revised the legend in Extended Data Table 2 to better explain this approach and avoid any confusion.

“4. Line 365: Why biome-level XGBoost models are needed where a global-generic model can already work on understanding factors of edge effects?”

Response 8: While the global model provides valuable insights into overarching patterns, it may not fully capture biome-specific variations in edge effects. Certain environmental factors may play a more significant role in different biomes, which is why we ran separate biome-level models. These models allowed us to assess whether the importance of each environmental factor varies across biomes. However, we acknowledge that the key biome-specific insights regarding the contribution of each environmental factor to edge effect variation were already addressed in the global model. To improve conciseness, we have removed the detailed discussion of the biome-level models and retained a single sentence that highlights the most important finding: that mean annual precipitation is the most important variable in the tropical biome (L167-168).

“5. Figure 2: Why there are different SHAP values in e-f compared to d? Same SHAP values should be presented regarding contributions from mean annual precipitation.”

Response 9: Thank you for your comment regarding the SHAP values in Figures 2d-f. We recognize that the original presentation may have led to some confusion regarding the SHAP values presented for mean annual precipitation (MAP) across these figures. In the original figures (2e-2f), we displayed only the interaction effects between MAP and other variables (MAT or Agriculture), with the “main effect” of MAP removed. Based on your suggestion, we have now revised the figures to present interaction effects alongside the contributions from MAP, as shown in Figures 2d and 2e. To avoid redundancy, we have removed the original SHAP dependence plot of MAP, as its information is already captured in these revised figures.

Reviewer #2

“Thank you for this interesting study. I really enjoyed reading your article, which is well structured and uses advanced statistical modelling to answer the proposed research questions. Apart from a few major comments (described below), I have also a number of text edits, additional literature and raised a few questions with respect to clarity of some parts. Overall, however, this is a very relevant and potentially impactful study which further underlines the detrimental interaction between habitat fragmentation and climate change. The authors call for a change in policy and management and for the consideration of edge effects, which is timely and aligns well with previous recent literature.”

Response 10: We sincerely appreciate your thoughtful and positive feedback on our study. We have carefully addressed your major comments and specific suggestions below and look forward to any further feedback.

“Larger comments:

The introduction is complete and addresses the state of the art and knowledge gaps profoundly. However, I was missing some potentially relevant literature at some points, which I added in my comments.”

Response 11: We appreciate your suggestions for additional literature. In response, we have incorporated “soil conditions” into the sentence (L25-27) as suggested. Additionally, we have included several of the recommended references in this sentence to further strengthen the introduction.

(e.g. Vanneste, T. *et al.* Trade-offs in biodiversity and ecosystem services between edges and interiors in European forests. *Nat Ecol Evol* **8**, 880–887 (2024).

Meeussen, C. *et al.* Microclimatic edge-to-interior gradients of European deciduous forests. *Agric For Meteorol* **311**, 108699 (2021).)

“The results section is quite extensive and some parts can be removed to increase the focus of the story line. I also noticed that the results are already intermixed with discussions. I actually like this way of writing, but then I found it somewhat strange that you have a separate discussion section. I suggest to choose one of both approaches, but potentially it is also limited by the journal's guidelines.”

Response 12: Thank you for your feedback regarding the structure of the Results and Discussion sections. In response, we have integrated the Discussion into the Results to create a more cohesive narrative, particularly around macroenvironmental drivers of edge effect patterns. Additionally, we have introduced a separate subsection titled “Comparison with previous studies” (L268) to specifically address how our findings relate to, and differ from, those of earlier field-based and large-scale studies—especially in temperate forests.

“The methodology is quite complex and I have to admit that I was not familiar with all modelling methods. However, I found that the authors described the rather advanced methodology in a relatively logical way, which was much appreciated. The only larger concern I had with the applied methodology was the use of relative “old” maps from Hansen *et al.* and Harris *et al.* I was wondering if there is not a more recent source available at a comparable resolution, and what effect this could have on your outcomes. I think the authors could better justify the use of these older maps and discuss this in the main text. Do you expect that the situation now is the same as in 2000, or could there be differences in your findings? Recent field-based measurements in 9-m radius plots from across Europe, for

instance, find an opposite pattern (higher biomass in the edges). Of course, the scale of these studies is entirely different, but still I think this requires some deeper discussion.”

Response 13: We appreciate your feedback and concerns regarding the use of datasets. We have now provided a clearer justification for our choice of data in the main text and methods section (L256-266 and L345-349). Specifically, we emphasize that the Harris et al. (2000) biomass map represents the best available global dataset at such a fine (30m) resolution, which is fundamental to our analysis, and we opted for the Hansen et al. (2000) forest cover data to ensure temporal consistency.

We also acknowledge that significant forest changes have occurred globally since 2000, which could influence edge effects if biomass gains or losses were unevenly distributed between edges and interiors. To address this, we have added a discussion in the main text highlighting how future studies could explore the sensitivity of our findings using more recent datasets.

“Finally, fine-scale drivers such as soil parameters or forest microclimate were not included here, which is fine because I do not think this data is already available at a global extent. Yet, these drivers, and especially microclimate, can vary quite markedly from edge to core and have a strong influence in forests. Maybe this could also be discussed a bit more.”

Response 14: We appreciate your insightful comment regarding the potential influence of fine-scale drivers, such as soil parameters and forest microclimate, on edge effects. We acknowledge that these factors can exhibit significant variation from edge to core and play a crucial role in shaping forest dynamics.

Our current global-scale analysis primarily focuses on broader environmental gradients (MAP, MAT, Agriculture, wind speed, soil moisture, elevation, and slope) on edge effect variations due to data availability at this scale. In our analysis, we used an Extreme Gradient Boosting (XGBoost) model to identify the primary drivers of variation in edge effects across the globe. While this approach effectively captured the influence of broader environmental gradients, it may have overlooked the significant contributions of fine-scale factors on edge effect variations.

As you point out, soil parameters and microclimate can have strong, localized effects. For instance, edge-induced changes in solar radiation, wind exposure, and humidity can significantly alter microclimate conditions, which in turn can influence tree physiology, growth, and mortality. Similarly, variations in soil nutrient availability, moisture content, and pH can affect plant community composition and ecosystem processes. Besides introducing these factors in our Introduction (L32-37), we have now further emphasized their importance in the discussion to highlight their potential role in shaping edge effect patterns (L204-216).

“Minor suggestions:

L7: I suggest to use: Direction and magnitude of edge effects”

Response 15: Done. (L7)

“L9: Please add for clarity: “on biomass” after edge effect”

Response 16: Done. (L9)

“L12: I suggest to use past tense: were instead of are”

Response 17: Done. (L12)

“L24: I think it is just km instead of km²?”

Response 18: The citation numbered “2” may have caused confusion with square kilometers. To clarify, we have revised the sentence (L23-25).

“L24: please move ‘ before comma”

Response 19: Done. (L25)

“L24-25: There are also edge-to-interior gradients in edaphic conditions We have actually very recently published some articles around these topics: <https://doi.org/10.1111/jvs.12844> <https://doi.org/10.1016/j.agrformet.2021.108699> <https://doi.org/10.1038/s41559-024-02335-6> These could potentially be relevant for the intro?”

Response 20: Thank you for pointing out the relevance of edge-to-interior gradients in edaphic conditions and for sharing those publications. We have incorporated your feedback by adding 'soil conditions' to the sentence (L25-27) and citing Vanneste et al. and Meeussen et al.

Vanneste, T. et al. Trade-offs in biodiversity and ecosystem services between edges and interiors in European forests. *Nat Ecol Evol* 8, 880–887 (2024).

Meeussen, C. et al. Microclimatic edge-to-interior gradients of European deciduous forests. *Agric For Meteorol* 311, 108699 (2021).

“L39: The additional literature that I have provided above, could actually also be very relevant here. It’s not global, but already a generalization across the European temperate biome.”

Response 21: We appreciate you highlighting the relevance of your provided literature to L39. After reviewing them, we determined that Vanneste et al. addressed edge effects on biomass, which is the specific focus of that sentence. We have now included it in our citations (L41).

“L61: direction instead of sign?”

Response 22: Done. (L66)

“L69: Hansen et al., I was just wondering if there is not a more recent source available as this map already dates from 2000? The EU provides global forest cover maps for 2020, but perhaps at a much coarser resolution (https://forobs.jrc.ec.europa.eu/GF_C#data_download). You also have the ESA world cover map (<https://esa-worldcover.org/en>), which includes forest cover. Maybe recent satellite or Lidar maps (I'm not an expert on this topic) could provide a better resolution and more recent view of the situation?”

Response 23: Thank you for your concern regarding the 2000 datasets. While newer forest cover maps exist, corresponding biomass data at this resolution is unavailable. As mentioned in *Response*

13, the Harris 2000 biomass map remains the most recent global dataset at 30m resolution. Therefore, for temporal consistency, we used the 2000 Hansen et al. and Harris et al. maps, as detailed in L256-266 and L345-349.

“L80: please add i.e. before negative (part between brackets)”

Response 24: Thank you for the suggestion. We have clarified the phrasing by explicitly writing out “negative edge effects” in the revised sentence (L84-86).

“L106: Variable or driver sounds more logical to me than “feature””

Response 25: We have implemented your suggestion and replaced all instances of 'feature' with 'variable' throughout the manuscript.

“L107: arranged instead of aligned?”

Response 26: We have revised the sentence. (L111-112)

“L115-118: I'm not sure that I understand this subfigure g completely. Is agriculture also included? Could you clarify?”

Response 27: Thank you for pointing out the need for clarification regarding subfigure f (previously g). This subfigure highlights only the most important variable per biome, and does not show all variables included in the models. Agriculture, among other variables, is included in the tropical/subtropical and temperate biome-level models. However, it was excluded from the boreal model due to high collinearity with MAT (Spearman correlation > 0.7), as detailed in the Methods section (L432-437). To further clarify this, we have added a sentence to the legend (L121-123).

“L128-130: I found this sentence rather complex with too many commas. Could you rephrase?”

Response 28: We appreciate your feedback. We have revised the sentence to improve clarity (L132-135).

“L148-154: There is already a lot of discussion in this part, intermixed with results. Of course this is fine but then I was confused that you have an additional discussion section below.”

“L156-163: also a lot of discussion”

“L165-174: discussion”

Response 29: Thank you for pointing this out. As noted in *Response 12*, we have removed the separate Discussion section and integrated these discussion elements into the Results to create a unified and coherent narrative.

“L185-193: I have the feeling these biome level distinctions were already discussed in the parts above. I wonder if it is really necessary to repeat it separately here.”

Response 30: Thank you for this helpful observation. We agree that the biome-level distinctions had already been sufficiently addressed earlier in the text. In response, we have removed the redundant paragraph to improve clarity and avoid repetition.

“L197: The figure legend in (a) seems to show Tg, but in (b) Pg is mentioned. Could you clarify?”

Response 31: Thank you for pointing out the discrepancy in units between Figure 3b and 3c. You are correct that Figure 3b displays AGB loss in Tg (teragrams), while Figure 3c shows it in Pg (petagrams). This difference is intentional. Figure 3b illustrates grid cell level AGB loss, while Figure 3c presents the sum of these grid cell level losses across entire biomes. To accurately represent the total AGB loss at the global and biome level, a higher unit (Pg) was more appropriate. We have clarified this distinction in the figure legend to prevent further confusion (L222 and L225).

“L213: 58 Pg, could you add uncertainty (eg SD)?”

Response 32: Thank you for your suggestion regarding the inclusion of uncertainty. We have now added uncertainty estimates, specifically 95% confidence intervals (CI), to the AGB loss values (L244-245 and L246-248). We have also updated Figure 3c accordingly. The methodology for estimating these uncertainties is detailed in the Methods section, where we explain how we calculated the 95% confidence intervals using standard error-based bounds and propagated them consistently through our analyses (L486-490).

“L229-231: I think this needs a bit deeper discussion. There is indeed a strong discrepancy between your findings and the ones by Meeussen et al. in *Science of the Total Environment*. Of course, I agree there is also a huge difference in scale, with Meeussen et al. using ground-based TLS to quantify plot-scale (9 m radius) AGB, while you are using globally gridded data. Still, Meeussen et al. also uses a continuous edge-interior gradient of 100m from different locations with a wide range of macroclimates and landscape contexts. They reported a consistent pattern of AGB being 95% higher in edges than interiors. So, I was wondering how the field-based data replicated at continental scale and global data still give opposite patterns. Could you elaborate?”

Response 33: Thank you for raising this important point and for highlighting the apparent discrepancy between our findings and those of Meeussen et al. (2021). We appreciate the opportunity to clarify this comparison in more depth.

At first glance, our results may seem to contrast with field-scale studies reporting higher biomass density near the edges of temperate forests (L269-271), such as Meeussen et al. (2021) in *Science of the Total Environment*, who reported a consistent 95% increase in aboveground biomass (AGB) near forest edges.

However, Meeussen et al. observed increased biomass only within the first 30 m of a forest edge (as seen in Fig. 3 of Meeussen et al. (2021)). This 30 m range of distance would be contained entirely within a single pixel for our 30 m resolution global analysis (L271-274). Furthermore, while Meeussen et al. investigated edge effects up to 100m, which is the distance they considered ‘interior’, in our global 30 m resolution analysis, a 100m distance from an edge may still remain within the edge zone, and not the interior.

Indeed, when calculating the maximum distance value within each grid cell, the average of maximum distance values across all grid cells was 547.33 m, which is more than five times deeper than 100 m. Additionally, the ‘depth of edge influence’ that divide edge and interior zones, which is the mean distance of points with 90% quantile of biomass, was 336 m, 826 m, 235 m, and 258 m in global,

tropical, temperate, and boreal scale (L236-237). This means that all forest regions within these distance values, which are all deeper than 100m, were considered 'edge zone'. Our global analysis, while capturing broader trends, may average out or obscure the strong, localized edge effects observed in Meeussen et al.'s study.

Thus, rather than conflicting with prior results, our findings suggest instead that the increases in temperate forest biomass near edges observed previously represent a fine-scale trend that is nested within a larger-scale, opposing macrotrend.

Further research will be necessary to investigate the ecological implications and contrasting impacts on carbon storage of these two patterns, which could vary significantly with the scale of analysis as well as a landscape's degree of forest fragmentation. Future research could explore the scaling of edge effects by combining field-based measurements with high-resolution remote sensing data. This could help to better understand how local edge effects translate to broader spatial scales (L274-279).

"L249-250: Please change to a less strong wording, e.g. is in line with or corroborates"

Response 34: We replaced "confirms the robustness" with "corroborates" (L300).

"L272-276: Very convincing conclusion!"

Response 35: Thank you for your positive feedback.

"L286-287: Is this the most recent source available? If yes, I would highlight that in the text and write that it is the best available product at this moment and at this fine resolution. What would you expect from using a more recent map? How could this impact your results? Maybe shortly elaborate on that, potentially even in the main text."

Response 36: As stated in *Response 13*, these concerns have been highlighted in L256-266 and L345-349.

"L304: describe"

Response 37: Correction done (L368)

"L321: mile, please consider SI units"

Response 38: We appreciate your suggestion regarding the use of SI units. To address this, we have converted mile to meters in the manuscript while maintaining consistency with the original FIA data specifications. The use of a 1-mile buffer was necessary because FIA plot locations have a built-in positional uncertainty within a 1-mile radius. We have added a sentence clarifying this (L384-387).

"L343: I have to be honest here, but I'm not at all an expert on this type of model. However, I feel that the authors have quite nicely explained the complex workflow, which was highly appreciated."

Response 39: Thank you for your feedback. We are glad to hear that our explanation was clear.

“L345-346: Could you briefly repeat the different tested drivers here for clarity? I was also wondering what the effect of small-scale driver would be, such as some soil parameters or even forest microclimate. I feel this requires a few lines of discussion in the main text. In our recent studies (<https://doi.org/10.1038/s41559-024-02335-6>) we found that these fine-scale drivers (esp. microclimate) can actually have quite strong effects.”

Response 40: Thank you for your suggestion. To improve clarity, we have added a sentence listing all tested drivers (L414-417). Regarding small-scale drivers, we have added a paragraph discussing their potential influence (L204-216), as stated in *Response 14*,

“L363: Please add equal to or = after RMSE (also for following values).”

Response 41: Done (L430)

“L366: Please add: separately after forests.”

Response 42: Done (L433)

“L372: Idem here. Please add equal signs before values for clarity.”

Response 43: Done (L439-441)

“L378: Feature, maybe this term is more related to the type of model you are using, but in my opinion "variable" sounds a bit more fitting. It's also what I'm used to from literature. I think variables is also the more commonly used term in traditional statistics.”

Response 44: We replaced all mentions of 'feature' with 'variable' throughout the manuscript.

“L396: “To do this”. I suggest to change this to: To this end”

Response 45: Done (L463)

“L402: “Edge limit”. In traditional forest edge literature, I think this is called "depth of the edge effect".”

Response 46: Thank you for your suggestion. We have replaced “edge limit” with “depth of edge influence” to align with traditional forest edge literature terminology (L470-473).

“L410: please change to “if the edge limit was equal to 200 m”.”

Response 47: Done (L480)

“L411: I suggest to change to “within 200 m from the nearest edge”.”

Response 48: Done (L480-481)

“L415: Maybe you could also convert this amount to carbon stocks in order to quantify the actual impact on the global C balance?”

Response 49: Thank you for your suggestion. In response, we have added a statement quantifying the implications of biomass loss for the global carbon balance (L250-253).

“I have also added my text edits and minor comments in the attached PDF file.
Good luck with the review! I already look forward to the edited version.”

Response 50: Thank you for your thorough review and for providing your text edits and comments. We appreciate your detailed feedback. We have implemented your suggestions and believe they have significantly improved the manuscript.

Reviewer #3

“This is an important paper at two scales, the scale of biomes and the scale of the globe. The global detail and rigor of this analysis brings into sharp focus the knock-on effects of habitat loss and fragmentation on ecosystems and carbon lost from ecological systems. Because such a high proportion of forest is near the edge, this study finds that the impacts of habitat fragmentation on carbon lost is massive. At the scale of biomes, this study resolves a pattern that has, in the past, proven inconsistent: variation in edge effects on aboveground biomass near, for example, edges of tropical versus temperate forests. Discrepancies have hindered understanding and conservation attention in this area. This new analysis, with unprecedented extent and (at that extent) precision points to the generality of the loss of biomass near habitat edges. With regard to edge effects, I'd like to see more depth in explanation of the following three questions.”

Response 51: We extend our sincere gratitude to the reviewer for positive and constructive feedback on our manuscript. We deeply appreciate the time and effort dedicated to evaluating our work and for posing such insightful questions. Our responses to the three points raised are provided below.

“1. What explains the “short” distance of edge effects? Working with data from tropical forests, Chaplin-Kramer, et al. (2019) that you cited shows edge effects that extend much deeper into the forest. (this paper deserves to be cited Broadbent, et al. 2008 see Fig. 6)”

Response 52: Thank you for your comment and for drawing attention to Broadbent et al. (2008). We would like to clarify that our study does not attempt to define a fixed or universal distance for edge effects. Instead, we developed a dynamic, grid cell-level metric—the ‘depth of edge influence’—to classify forest pixels as either ‘edge’ or ‘interior’. This threshold was derived by identifying the mean distance to the edge for points in the 90th percentile of biomass density, following a logic similar to Chaplin-Kramer et al. (2015), who defined the depth at which 90% of asymptotic biomass is reached (L464-473).

This metric is intended to serve as a flexible boundary that adjusts to local biomass distributions. It allows consistent edge classification across diverse biomes. We have now added a figure (Fig. 3a) showing this depth of edge influence.

Based on this method, we found the average edge influence depths to be approximately 336 m globally, 826 m in tropical forests, 235 m in temperate forests, and 255 m in boreal forests (L236-237). These values—particularly for the tropics—align well with those reported in Chaplin-Kramer et al. (2015). We also tested alternative thresholds (Extended Data Fig. 12) and found our results to be robust. We hope this explanation addresses your concern regarding the seemingly ‘short’ edge distances.

Chaplin-Kramer, R. et al. Degradation in carbon stocks near tropical forest edges. *Nat Commun* 6, (2015).

“2. What explains the difference between these results of consistent negative edge effects across biomes, and other studies have found a different response? I found the explanations given to be interesting, but superficial. In fact, these and past results are consistent in one respect: the edge effect on biomass is much lower in the temperate than in tropical forest; perhaps this change in magnitude has been confused for change in direction of response?”

Response 53: Thank you for your valuable insight on the discrepancy of results across studies. We agree that some past findings, particularly in temperate forests, report positive edge effects on biomass (L269-271). However, these findings often come from fine-scale studies. For example,

Meeussen et al. (2021) observed increases in biomass near temperate forest edges—but only within the first 30 m from the edge, and with the deepest ‘interior’ defined as 100 m from the edge. In contrast, our 30 m resolution global dataset may not detect these fine-scale dynamics (L271-274), and may consider forests within 100 m still within the edge zone, not interior. Consequently, with our analysis spanning up to ~547 m per grid cell and a global average depth of edge influence of 336 m (L236-237), many areas labeled ‘interior’ in fine-scale studies fall within our definition of edge-affected zones.

As a result, localized biomass increases near forest edges may be real, but they occur within a broader pattern of biomass decline that emerges at landscape and biome scales. Rather than contradicting earlier findings, our results suggest that these fine-scale edge enhancements are nested within larger-scale negative trends. This likely reflects differences in the dominant drivers at each scale—for example, global climate gradients and disturbance regimes versus local microclimate buffering and species interactions.

We also note that classification thresholds (e.g., 10% tree cover in Morreale et al. (2021) vs. 30% in our study) can affect biomass estimates and edge interpretations. We have clarified these distinctions in the revised manuscript and agree that further work is needed to reconcile results across scales and methodological definitions (L274-279, L282-286, and L300-305). Integrating high-resolution remote sensing with field data could help bridge this gap in future research.

Meeussen, C. et al. Drivers of carbon stocks in forest edges across Europe. *Science of the Total Environment* 759, (2021).

Morreale, L. L., Thompson, J. R., Tang, X., Reinmann, A. B. & Hutyra, L. R. Elevated growth and biomass along temperate forest edges. *Nat Commun* 12, (2021).

“3. What is the mechanism of temperature effect on reduced aboveground biomass? I ask because there are two possible temperature effects that may be interacting. The authors focus on climate change. Yet, it is also well known that edges themselves affect (and can increase) temperatures (ie, Tuff, et al. 2016). What is the impact of environmental versus edge-caused temperature change? I do not work with remotely sensed data, and especially not at regional and global scales. Thus, I am not able to evaluate analyses in this paper. However, I have collaborated with such scientists and the data are appropriate for such global analyses.

Tuff, K.T., Tuff, T. and Davies, K.F., 2016. A framework for integrating thermal biology into fragmentation research. *Ecology letters*, 19(4), pp.361-374.

Broadbent, E.N., Asner, G.P., Keller, M., Knapp, D.E., Oliveira, P.J. and Silva, J.N., 2008. Forest fragmentation and edge effects from deforestation and selective logging in the Brazilian Amazon. *Biological conservation*, 141(7), pp.1745-1757.”

Response 54: Thank you for this thoughtful question on the temperature effects. We agree that both regional climate warming and localized edge-induced temperature increases contribute to reductions in aboveground biomass, and distinguishing their individual effects remains challenging—particularly at global scales.

Our results suggest that these two temperature drivers act synergistically. For tropical and temperate forests, climate warming raises overall average temperatures, while forest edges experience additional heating due to increased solar radiation exposure. This combined effect can push trees beyond their physiological tolerance limits, leading to increased heat stress, reduced growth, and higher mortality (L310-314).

In boreal forests, warming might initially benefit productivity in regions where temperature is a growth-limiting factor. However, this biome is experiencing disproportionately rapid climate change, and

conditions may soon exceed the adaptive capacity of many species (L315–322). Our results also show that boreal forests already experience significant relative biomass density losses due to edge effects compared to other biomes (Fig. 3c and Extended Data Fig. 11). Furthermore, climate warming is increasing the frequency and intensity of disturbances such as wildfires, insect outbreaks, and drought, which disproportionately impact forest edges due to their drier and more exposed conditions. These increased disturbances can lead to biomass loss.

Additionally, our results suggest that lower biomass near edges is associated with reduced canopy cover, as shown in our tree cover analysis (Extended Data Fig. 3). Reduced canopy cover near edges weakens the forest's microclimatic buffering effect, making edge environments more susceptible to temperature extremes (discussed in L209-213). While dense interior forests experience some mitigation of climate warming due to microclimate buffering, edges with lower canopy cover experience intensified heating effects. This diminished buffering could amplify the impact of climate warming on biomass, making edge environments particularly vulnerable to rising temperatures.

Submission ID NATECOLEVOL-24123528A

Title: A globally consistent negative effect of edge on aboveground forest biomass

Journal: Nature Ecology and Evolution

Reviewer #1

“The authors addressed the concerns I wrote about in my previous comments. I agree that a more concrete understanding of the mechanism can be encouraged for future work, and the authors included this in the discussion. It's also encouraging to see the code and generated data on forest edge distance, loss of biomass, and sensitivity to distance in the final publication. I recommend that the authors also share the full data processing scripts from start to finish, which would benefit the community.”

Response 1: Thank you very much for your positive follow-up review and for recognizing the improvements we made in response to your earlier comments. We are grateful for your suggestion regarding the data and data processing scripts. We will include a data and code repository alongside the final publication. We appreciate your encouragement and are happy to contribute to open science in this way.

Reviewer #2

“In my opinion, the authors have done an exceptionally good job providing highly adequate and convincing responses to the concerns raised in my previous review. Therefore, my remaining comments are only minor (I supply them as in-text edits in the PDF file attached). Hopefully, you find these remaining comments helpful and after implementation, I think this study is completely ready for publication.

After reading this revised version and the convincing reasoning in the rebuttal letter, I now understand why the 2000 maps were used and not a more recent (real-time) estimation of the forest cover and associated biomass situation. Yet, I still wonder a bit what the impact of a more recent map would be on your results. Will the situation have aggravated and differences between edge and interior biomass become even more pronounced? Or, at least for the situation here in Flanders and probably elsewhere in central Europe, forests have become so heavily fragmented that interiors virtually not exist any longer. Almost everything would be quantified as edge from a fine-scale (e.g., microclimate) point of view. In other words, differences between edges and edge-like interiors would not be much different.

Anyway, perhaps this goes to far and is more applicable to finer scales whereas your analyses are global. Good luck with the final revisions and already a big congratulations on the impressive work. Best regards,

Thomas Vanneste”

Response 2: Thank you sincerely for your thoughtful and supportive review. We greatly appreciate your recognition of the revisions made and your generous comments about the clarity and strength of our responses.

We're especially grateful for your insightful reflection on how forest fragmentation and the disappearance of true interiors in highly fragmented landscapes—such as in Flanders and parts of central Europe—might affect the distinction between edge and interior zones.

As noted in the revised manuscript (L238–240), we explained that our analysis relies on the year-2000 biomass dataset because it remains the highest-resolution (30 m) global product currently available. However, we fully agree that considerable forest change has occurred since then—especially in regions such as Flanders and central Europe where forest fragmentation has intensified. As you point out, in many such areas, true forest interiors have become extremely rare, meaning that almost all forest may now exhibit edge-like conditions from a microclimatic perspective.

This has important implications. With so little remaining interior, the edge–interior biomass gradient becomes increasingly blurred, not necessarily because edge effects have weakened, but because the reference baseline (interior forest) is itself degraded. Thus, the measurable contrast between edge and interior may appear smaller even though the landscape-wide biomass loss could actually be greater due to increased edge exposure.

In parallel, continued tropical deforestation since 2000 has likely intensified edge exposure and reduced intact interiors, especially in the Amazon and Southeast Asia. This would likely amplify total biomass losses, even if gradients in heavily impacted areas become less distinct. These changes could shift both the magnitude and distribution of edge effects in complex ways.

We also recognize that deforestation and reforestation can affect edges and interiors differently. Forest loss often begins at edges, which are more exposed and vulnerable, leading to faster biomass decline near those boundaries. Conversely, regrowth may occur more slowly near edges due to harsher conditions. These uneven patterns could lead to greater overall biomass change than what edge–interior gradients alone might suggest.

While our analysis captures the global pattern as of 2000, we agree that future studies using more recent datasets could help reveal how edge effects are evolving—and whether they are becoming stronger, weaker, or simply more difficult to detect due to fragmentation. We have expanded the discussion in the manuscript to reflect these points (L244-249). Thank you again for your very generous support.

Reviewer #3

“The authors have successfully addressed my and other reviewers' comments.”

Response 3: Thank you for your follow-up review and for confirming that your previous concerns have been fully addressed. We are grateful for your engagement and support throughout the review process.